# Incorporating sparse labels into hidden Markov models using weighted likelihoods improves accuracy and interpretability in biologging studies

**Evan Sidrow**[1]*, **Nancy Heckman**[1], **Tess M. McRae**[2], **Beth L. Volpov**[2], **Andrew W. Trites**[2,3], **Sarah M.E. Fortune**[2,4], **Marie Auger-Méthé**[1,2]

**1** Department of Statistics, University of British Columbia, Vancouver, British Columbia, Canada, **2** Institute for the Oceans and Fisheries, University of British Columbia, Vancouver, British Columbia, Canada, **3** Department of Zoology, University of British Columbia, Vancouver, British Columbia, Canada, **4** Department of Oceanography, Dalhousie University, Halifax, Nova Scotia, Canada

\* evan.sidrow@stat.ubc.ca

**Data availability statement:** All code and data are available at https://github.com/evsi8432/PHMM/.

## Abstract

Ecologists often use a hidden Markov model to decode a latent process, such as a sequence of an animal's behaviours, from an observed biologging time series. Modern technological devices such as video recorders and drones now allow researchers to directly observe an animal's behaviour. Using these observations as labels of the latent process can improve a hidden Markov model's accuracy when decoding the latent process. However, many wild animals are observed infrequently. Including such rare labels often has a negligible influence on parameter estimates, which in turn does not meaningfully improve the accuracy of the decoded latent process. We introduce a weighted likelihood approach that increases the relative influence of labelled observations. We use this approach to develop hidden Markov models to decode the foraging behaviour of killer whales (*Orcinus orca*) off the coast of British Columbia, Canada. Using cross-validated evaluation metrics and a detailed simulation study, we show that our weighted likelihood approach produces more accurate and understandable decoded latent processes compared to existing hidden Markov models and single-frame machine learning methods. Thus, our method effectively leverages sparse labels to enhance researchers' ability to accurately decode hidden processes across various fields.

## Introduction

The hidden Markov model, or HMM, is a common statistical model that is increasingly being used to understand the movements and behaviours of animals [1–3]. An HMM is a generalization of a mixture model that is used to decode a latent process of interest (e.g., a sequence of animal behaviours) from an observed time series (e.g., biologging data from tags attached to the animal). They have been used to uncover a wide variety of animal behaviours, including foraging activity [4,5] and habitat selection [6].

**Funding:** ES thanks the University of British Columbia and the Four-Year Doctoral Fellowship program for its support. MAM thanks the BC Knowledge Development Fund and the Canada Foundation for Innovation's John R. Evans Leaders Fund under grant 37715. MAM acknowledges the support of the Natural Sciences and Engineering Research Council of Canada (NSERC), Discovery grant RGPIN-2017-03867. MAM also thanks the Canadian Research Chairs program for Statistical Ecology. NH acknowledges the support of the Natural Sciences and Engineering Research Council of Canada (NSERC), Discovery grant RGPIN-2020-04629. We thank the Canadian Statistical Sciences Institute (CANSSI) for its support. We also acknowledge the support of the Natural Sciences and Engineering Research Council of Canada (NSERC) as well as the support of Fisheries and Oceans Canada (DFO). This project was supported in part by a financial contribution from the DFO and NSERC (Whale Science for Tomorrow). These sponsors did not play any role in the study design, data collection and analysis, decision to publish, or preparation of this manuscript.

**Competing interests:** The authors have declared that no competing interests exist.

Many ecological studies employ *unsupervised* HMMs, meaning that the true behaviours of the study animals are never directly observed and instead are predicted entirely from biologging data [7–10]. However, ecologists are often interested in predicting complicated animal behaviours (e.g., successful prey captures) that are difficult to identify from movement data alone [11]. For these behaviours, the relationship between an animal's behaviour and its movement is so complex that it is rarely fully characterized by a statistical model.

Foraging behaviour can be especially rare and difficult to identify, but it is often of prime interest in ecology [12,13]. For example, understanding foraging behaviour is vital for the conservation of northern and southern resident killer whales (*Orcinus orca*) off the coast of British Columbia [4,14,15]. Although both sub-populations have dietary and spatial overlap, northern residents (threatened) have a positive growth trajectory compared to southern residents (endangered) [15,16]. Studies have shown that various factors contribute to these population trends, including prey availability, pollutants, and vessel disturbances, but the exact causal mechanisms are not fully understood [4,14,17]. Each of these factors affects foraging ecology differently, so understanding how often and how successfully these sub-populations hunt may help explain differences in their population trajectories [15,18].

One solution to better identify animal behaviour is to fully observe and incorporate the animal's behaviours into the underlying model, in which case the HMM is *fully supervised*. Krogh et al. [19] showed that fully supervised HMMs can exhibit better predictive performance than unsupervised HMMs for gene finding, and fully supervised HMMs are used in fields ranging from speech recognition to medicine [20,21]. To our knowledge fully supervised HMMs are rare in ecology, but some animal behaviour studies use other fully supervised machine learning techniques [22,23]. However, these studies often focus on captive animals that are much easier to continuously observe compared to wild animals.

While fully observing an animal's behaviour in the wild can be prohibitively difficult or expensive, many ecological studies have behavioural information for a small subset of time. Occasional observations of an animal's behaviour can be incorporated into a *semi-supervised* HMM, and some notable ecological studies have used semi-supervised HMMs. For example, McClintock et al. [24] labelled a subset of hidden behavioural states of a grey seal (*Halichoerus grypus*) using its proximity to known "haul-out" and foraging sites. Alternatively, Pirotta et al. [9] assumed that northern fulmars (*Fulmarus glacialis*) begin every journey in some known behavioural state. Other studies that used semi-supervised HMMs include McRae et al. [25], who used drone footage to directly label the behaviour of killer whales for a subset of observation times, and Saldanha et al. [13], who used a multi-sensor approach to derive behavioural labels for red-billed tropicbirds (*Phaethon aethereus*). All of these studies demonstrate that incorporating partial labels into an ecological HMM can significantly improve its performance.

While behavioural labels can improve an HMM's prediction accuracy, many ecological studies only have access to labels for a small proportion of observations (e.g., <10%) [13,25]. In these cases, the labelled data often do not meaningfully affect the parameter estimates of an HMM because the likelihood is dominated by the unlabelled data [26,27]. A study by Ji et al. [28] used a weighted likelihood approach to increase the influence of labelled examples, but it assumed that labels correspond to independent time series. This approach does not apply when labels occur *within* a time series, which is often the case for ecological studies [13,25].

We introduce a novel weighted semi-supervised learning approach for hidden Markov models that allows practitioners to adjust the influence of sparse labels within a time series. We first review the definition of an HMM and current semi-supervised learning techniques for mixture models that lack time dependence. We then formalize a *partially hidden Markov*

*model*, or PHMM, which is designed to account for time series that are partially labelled, before introducing a weighted likelihood approach to balance the influence of labelled and unlabelled data within the model. Next, we present two case studies that use labels derived from video data and our weighted likelihood approach to achieve higher cross-validated accuracy compared to traditional HMMs and single-frame machine learning methods. Finally, we conduct a simulation study to investigate how different data regimes affect PHMM performance and guide practitioners when deciding how heavily to weight the likelihood of a PHMM.

## Background

### Hidden Markov models

Hidden Markov models describe time series that exhibit state-switching behaviour. They model an observed time series of length $T$, $\mathbf{Y} = \{Y_t\}_{t=1}^T$, by assuming that each observation $Y_t$ is generated from an unobserved hidden state $X_t \in \{1, \dots, N\}$. The hidden states $X_t$ are discrete and the observations $Y_t$ are usually (but not always) continuous. The sequence of all hidden states $\mathbf{X} = \{X_t\}_{t=1}^T$ is modelled as a Markov chain. The unconditional distribution of $X_1$ is denoted by the row-vector

$$\boldsymbol{\delta} = \begin{pmatrix} \delta^{(1)} & \cdots & \delta^{(N)} \end{pmatrix}, \tag{1}$$

where $\delta^{(i)} = \mathbb{P}(X_1 = i)$. Further, the distribution of $X_t$ given $X_{t-1}$ for $t = 2, \dots, T$ is denoted by the $N$-by-$N$ transition probability matrix

$$\boldsymbol{\Gamma}_t = \begin{pmatrix} \Gamma_t^{(1,1)} & \cdots & \Gamma_t^{(1,N)} \\ \vdots & \ddots & \vdots \\ \Gamma_t^{(N,1)} & \cdots & \Gamma_t^{(N,N)} \end{pmatrix}, \tag{2}$$

where $\Gamma_t^{(i,j)} = \mathbb{P}(X_t = j \mid X_{t-1} = i)$. For simplicity, we assume that $\boldsymbol{\Gamma}_t$ does not change over time (i.e. $\boldsymbol{\Gamma}_t = \boldsymbol{\Gamma}$ for all $t$) unless stated otherwise.

Each observation $Y_t$ is a random variable, where $Y_t$ given all other observations ($\mathbf{Y} \setminus \{Y_t\}$) and hidden states ($\mathbf{X}$) depends only on $X_t$. If $X_t = i$, then the conditional density or probability mass function of $Y_t$ is $f^{(i)}(\cdot; \theta^{(i)})$, where $\theta^{(i)}$ are the parameters describing the state-dependent distribution of $Y_t$. The collection of all state-dependent parameters is $\boldsymbol{\theta} = \{\theta^{(i)}\}_{i=1}^N$. The probability density of $\mathbf{Y}$, following an HMM with initial distribution $\boldsymbol{\delta}$, transition matrix $\boldsymbol{\Gamma}$, and state-dependent parameters $\boldsymbol{\theta}$, evaluated at $\mathbf{y} = \{y_t\}_{t=1}^T$, is

$$p(\mathbf{y}; \boldsymbol{\delta}, \boldsymbol{\Gamma}, \boldsymbol{\theta}) = \boldsymbol{\delta} P(y_1; \boldsymbol{\theta}) \prod_{t=2}^T \boldsymbol{\Gamma} P(y_t; \boldsymbol{\theta}) \mathbf{1}_N^\top, \tag{3}$$

where $\mathbf{1}_N$ is an $N$-dimensional row vector of ones and $P(y_t; \boldsymbol{\theta})$ is an $N \times N$ diagonal matrix with entry $(i,i)$ equal to $f^{(i)}(y_t; \theta^{(i)})$. Parameter estimation for HMMs often involves maximizing Eq (3) with respect to $\boldsymbol{\delta}$, $\boldsymbol{\Gamma}$, and $\boldsymbol{\theta}$. Fig 1 shows an HMM as a graphical model. For a more complete introduction to HMMs, see Zucchini et al. [29].

### Semi-supervised mixture models

Semi-supervised learning is a paradigm in machine learning that harnesses both labelled and unlabelled data to enhance model performance [26]. There is a large taxonomy of semi-supervised learning techniques, but here we focus on generative mixture models because an

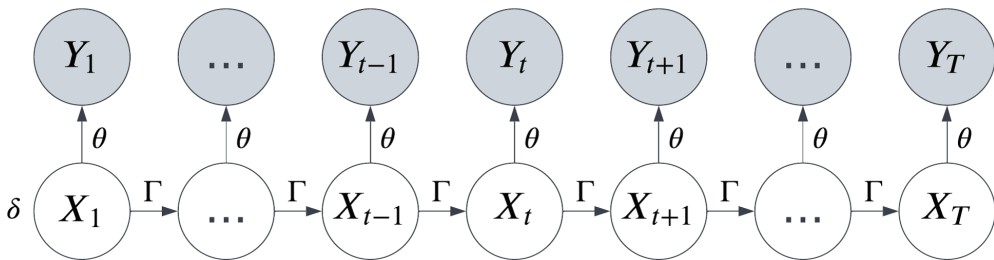

**Fig 1. Graphical representation of an HMM.** $X_t$ corresponds to an unobserved latent state at time $t$ whose distribution is described by a Markov chain. $Y_t$ corresponds to an observation at time $t$, where $Y_t$ given all other observations $\mathbf{Y} \setminus \{Y_t\}$ and hidden states $\mathbf{X}$ depends only on $X_t$.

HMM is a generalization of a mixture model that includes serial dependence between its hidden states [30,31]. Unfortunately, many semi-supervised learning techniques for mixture models do not account for the time dependence of HMMs. As such, we build off of current approaches for mixture models and develop a novel semi-supervised learning technique for HMMs.

A mixture model is a simpler version of a hidden Markov model where the hidden states $\mathbf{X} = \{X_t\}_{t=1}^T$ are modelled as independent categorical random variables instead of a Markov chain. The distribution of $X_t$ is denoted by the row-vector

$$\boldsymbol{\pi} = \begin{pmatrix} \pi^{(1)} & \cdots & \pi^{(N)} \end{pmatrix}, \ \forall \, t = 1, \dots, N,$$

where $\pi^{(i)} = \mathbb{P}(X_t = i)$. A sequence of observations $\mathbf{y} = \{y_t\}_{t=1}^T$ then has the probability density function

$$p(\mathbf{y} \, ; \, \boldsymbol{\pi}, \boldsymbol{\theta}) = \prod_{t=1}^T \left( \sum_{i=1}^N \pi^{(i)} f^{(i)}(y_t; \theta^{(i)}) \right). \tag{4}$$

Now, suppose that a subset of time indices $\mathcal{T} \subseteq \{1, \dots, T\}$ have corresponding labels $\mathbf{Z} = \{Z_t\}_{t \in \mathcal{T}}$. Labels are often observed at random times (e.g., aerial drones observe whale behaviours at random times), but we assume that $\mathcal{T}$ is fixed, as is common for many semi-supervised learning techniques [26,32]. Like $Y_t$, each label $Z_t$ is a random variable generated from its corresponding hidden state $X_t$. The state space of $Z_t$ is general, but for simplicity we assume that $Z_t \in \{1, \dots, N\}$. Given all other labels ($\mathbf{Z} \setminus \{Z_t\}$), observations ($\mathbf{Y}$), and hidden states ($\mathbf{X}$), we assume that $Z_t$ depends only on $X_t$ for each $t \in \mathcal{T}$. If $X_t = i$, then the label $Z_t$ has probability mass function $g^{(i)}(\cdot; \beta^{(i)})$, with parameters $\beta^{(i)}$. Denote a fixed realization of labels $\mathbf{Z}$ as $\mathbf{z} = \{z_t\}_{t \in \mathcal{T}}$. Then, the joint probability density of $\mathbf{y}$ and $\mathbf{z}$ for semi-supervised mixture models is

$$p(\mathbf{y}, \mathbf{z} \, ; \, \boldsymbol{\pi}, \boldsymbol{\theta}, \boldsymbol{\beta}) = \prod_{t \in \mathcal{T}} \left( \sum_{i=1}^N \pi^{(i)} f^{(i)}(y_t; \theta^{(i)}) g^{(i)}(z_t; \beta^{(i)}) \right) \prod_{t \notin \mathcal{T}} \left( \sum_{i=1}^N \pi^{(i)} f^{(i)}(y_t; \theta^{(i)}) \right), \tag{5}$$

where $\boldsymbol{\beta} = \{\beta^{(i)}\}_{i=1}^N$. To write Eq (5) in a simpler form, we define $z_t = \varnothing$ for all unlabelled observations (i.e., for all $t \notin \mathcal{T}$) and set $g^{(i)}(\varnothing; \beta^{(i)}) = 1$. This abuse of notation results in a

relatively simple probability density function for semi-supervised mixture models:

$$p(\mathbf{y}, \mathbf{z} \; ; \; \boldsymbol{\pi}, \boldsymbol{\theta}, \boldsymbol{\beta}) = \prod_{t=1}^{T} \left( \sum_{i=1}^{N} \pi^{(i)} f^{(i)}(y_t; \theta^{(i)}) g^{(i)}(z_t; \beta^{(i)}) \right). \tag{6}$$

Eq (6) can be used to construct a likelihood and maximized with respect to $\boldsymbol{\pi}$, $\boldsymbol{\theta}$, and $\boldsymbol{\beta}$ to perform semi-supervised inference on mixture models [26]. In some scenarios, subject matter experts can identify the labels $\mathbf{z}$ with certainty. In this case, $Z_t = X_t$ for all $t \in \mathcal{T}$, the parameters $\boldsymbol{\beta}$ do not need to be inferred, and $g^{(i)}$ takes the form

$$g^{(i)}(z_t) = \begin{cases} 1, & z_t \in \{i, \varnothing\}, \\ 0, & z_t \notin \{i, \varnothing\}. \end{cases} \tag{7}$$

This formulation of $g^{(i)}$ implies that, for all labelled observations, if the hidden state $X_t$ is equal to $i$, then the corresponding label $Z_t$ also equals $i$ with probability 1. We define $g^{(i)}$ as in Eq (7) in our case studies. However, if subject matter experts are not confident in their labels, or if a hidden state $X_t$ could generate one of multiple labels $Z_t$, we recommend parameterizing $g^{(i)}$ and inferring the parameters $\boldsymbol{\beta}$.

## Weighted likelihood for semi-supervised mixture models

One issue in semi-supervised learning occurs when the number of observations $T$ is much larger than the number of labels $|\mathcal{T}|$. In this case, the labelled data do not meaningfully affect maximum likelihood parameter estimates [26]. As a solution, Chapelle et al. [26] introduce a parameter $\lambda \in [0, 1]$ which represents the relative weight given to unlabelled observations. In particular, they define a weighted likelihood $\tilde{\mathcal{L}}_\lambda$ based on Eq (6) with weights $\tilde{w}_\lambda$ as follows:

$$\tilde{w}_\lambda(z_t) = \begin{cases} (1 - \lambda) \frac{T}{|\mathcal{T}|}, & z_t \in \{1, \dots, N\} \\ \lambda \frac{T}{T - |\mathcal{T}|}, & z_t = \varnothing \end{cases}, \tag{8}$$

$$\tilde{\mathcal{L}}_\lambda(\boldsymbol{\pi}, \boldsymbol{\theta}, \boldsymbol{\beta} \; ; \; \mathbf{y}, \mathbf{z}) = \prod_{t=1}^{T} \left( \sum_{i=1}^{N} \pi^{(i)} f^{(i)}(y_t; \theta^{(i)}) g^{(i)}(z_t; \beta^{(i)}) \right)^{\tilde{w}_\lambda(z_t)}. \tag{9}$$

Using this formulation, setting $\lambda = 0$ removes all unlabelled data, setting $\lambda = 1$ removes all labelled data, and setting $\lambda = (T - |\mathcal{T}|)/T$ returns a likelihood that corresponds to the joint density from Eq (6). It is unlikely that a practitioner would prefer setting $\lambda > (T - |\mathcal{T}|)/T$, as this weights unlabelled observations more heavily than labelled observations. In practice, researchers often select $\lambda$ by performing cross validation with an appropriate model evaluation metric [26].

The weighted likelihood $\tilde{\mathcal{L}}_\lambda$ is a specific instance of a much more general class of relevance-weighted likelihoods that has been studied extensively. Hu et al. [32] provide a comprehensive review of weighted likelihoods. Under their paradigm, the probability density of the labelled data $\{Y_t, Z_t\}_{t \in \mathcal{T}}$ is given by Eq (5), but the probability density of the unlabelled data $\{Y_t\}_{t \notin \mathcal{T}}$ is some unknown density that "resembles" the density of the labelled data in some sense. In particular, Hu et al. [32] formally define the notion of 'resemblance' using Boltzman's entropy, and the weight $\lambda$ corresponds to how much the density of the unlabelled data resembles the density of the labelled data under this definition. Hu et al. [33] prove the consistency and asymptotic normality of maximum weighted likelihood estimators under

certain regularity conditions. The relevance-weighted likelihood literature thus gives useful theoretical guarantees related to the weighted likelihood for mixture models. Unfortunately, these guarantees usually assume that the observations $\mathbf{y}$ are independent, which is not true for HMMs.

## Weighted likelihood for semi-supervised learning in hidden Markov models

Our weighted likelihood approach for semi-supervised learning in HMMs begins by writing down the probability density associated with a partially observed HMM. We use the same notation as described above, namely random labels $\mathbf{Z} = \{Z_t\}_{t \in \mathcal{T}}$, where $Z_t$ is generated from hidden state $X_t$ and, conditioned on $\mathbf{X}$, $\mathbf{Y}$, and $\mathbf{Z} \setminus \{Z_t\}$, $Z_t$ depends only on $X_t$. As before, a fixed realization of $\mathbf{Z}$ is denoted as $\mathbf{z} = \{z_t\}_{t \in \mathcal{T}}$, and we abuse notation by setting $z_t = \varnothing$ for all unlabelled observations (i.e., $t \notin \mathcal{T}$) and $g^{(i)}(\varnothing; \beta^{(i)}) = 1$. The joint density of the observations $\mathbf{y}$ and labels $\mathbf{z}$ for an HMM is thus

$$p(\mathbf{y}, \mathbf{z} \,;\, \boldsymbol{\delta}, \boldsymbol{\Gamma}, \boldsymbol{\theta}, \boldsymbol{\beta}) = \boldsymbol{\delta} P(y_1, z_1; \boldsymbol{\theta}, \boldsymbol{\beta}) \prod_{t=2}^{T} \boldsymbol{\Gamma} P(y_t, z_t; \boldsymbol{\theta}, \boldsymbol{\beta}) \mathbf{1}_N^\top, \tag{10}$$

where $P_t(y_t, z_t; \boldsymbol{\theta}, \boldsymbol{\beta})$ is an $N \times N$ diagonal matrix where entry $(i,i)$ is $f^{(i)}(y_t; \theta^{(i)}) g^{(i)}(z_t; \beta^{(i)})$. We refer to this model as a *partially hidden Markov model*, or PHMM.

Incorporating partial labels in an HMM to define a PHMM is relatively straightforward, but defining a weighted likelihood for PHMMs is more complicated. Recall that each term in Eq (9) is a scalar value raised to the power of some weight. However, each term in Eq (10) is a matrix, so it is not straightforward to raise each term to the power of a (possibly fractional) weight. While it is possible to calculate fractional powers of matrices, doing so can be computationally expensive and the result can be difficult to interpret [34]. Alternatively, Hu et al. [35] derive a relevance-weighted likelihood for dependent data using the same paradigm as Hu et al. [32]. Although their method is broadly applicable, it does not apply to HMMs. Namely, they adopt a paradigm where each observation $Y_t$ has a corresponding set of parameters $\{\boldsymbol{\delta}_t, \boldsymbol{\Gamma}_t, \boldsymbol{\theta}_t\}$, and they assume that $Y_t$ depends only on its corresponding parameters and the previous observations $\{Y_{t'}\}_{t'=1}^{t-1}$. However, this assumption is violated for an HMM because $Y_t$ depends on $X_t$, which in turn depends upon the previous parameters $\{\boldsymbol{\delta}_{t'}, \boldsymbol{\Gamma}_{t'}, \boldsymbol{\theta}_{t'}\}_{t'=1}^{t-1}$. See Eq (3) of Hu et al. [35] for more details.

A weighted likelihood for PHMMs should have three desired properties. First, the weighted likelihood should reduce to Eq (10) for some "natural" weight, just as $\lambda = (T - |\mathcal{T}|)/T$ does for the weighted likelihood in Eq (9). Second, some weight should correspond to ignoring all unlabelled data, just as $\lambda = 0$ does for the weighted likelihood in Eq (9). These two properties allow practitioners to intuitively select a weight that balances a natural weighting scheme with one that completely ignores all unlabelled data. Finally, the weighted likelihood should be relatively simple and intuitive compared to the standard likelihood from Eq (10). We thus propose the weighting parameter $\alpha \in [0, 1]$ and the following weighted likelihood for partially hidden Markov models:

$$w_\alpha(z_t) = \begin{cases} 1, & z_t \in \{1, \ldots, N\} \\ \alpha, & z_t = \varnothing \end{cases}, \tag{11}$$

$$\mathcal{L}_\alpha(\boldsymbol{\delta}, \boldsymbol{\Gamma}, \boldsymbol{\theta}, \boldsymbol{\beta} \,;\, \mathbf{y}, \mathbf{z}) = \boldsymbol{\delta} P(y_1, z_1; \boldsymbol{\theta}, \boldsymbol{\beta})^{w_\alpha(z_1)} \prod_{t=2}^{T} \boldsymbol{\Gamma} P(y_t, z_t; \boldsymbol{\theta}, \boldsymbol{\beta})^{w_\alpha(z_t)} \mathbf{1}_N^\top, \tag{12}$$

This formulation satisfies the three desired properties listed above. First, if $\alpha = 0$, then the term corresponding to an unlabelled observation $t$ is $\mathbf{\Gamma} P(y_t, z_t; \theta, \beta)^0 = \mathbf{\Gamma}$. Therefore, the likelihood of a PHMM with $\alpha = 0$ is identical to the likelihood of an HMM that treats all unlabelled observations as totally missing [29]. Next, if $\alpha = 1$, then the term corresponding to an unlabelled observation $t$ is $\mathbf{\Gamma} P(y_t, z_t; \theta, \beta)^1 = \mathbf{\Gamma} P(y_t, z_t; \theta, \beta)$. In this case, the weighted PHMM likelihood in Eq (12) corresponds to the standard PHMM density in Eq (10). Finally, we argue that this formulation is intuitive, as it weights unlabelled observations using some power of $\alpha$ and leaves labelled observations unaltered. Fig 2 shows graphical representations of PHMMs for several values of $\alpha$ with only $Z_1$, $Z_{t-1}$, and $Z_{t+1}$ observed for some fixed $t$.

Thus far we have examined the likelihood $\mathcal{L}_\alpha$ of a single time series, but in many practical applications multiple independent time series are observed from the same process (e.g., multiple killer whale tag deployments). In this case, we denote the total number of time series by $S$ and index them with $s$. Then, $\mathbf{y}_s = \{y_{s,t}\}_{t=1}^{T_s}$ and $\mathbf{z}_s = \{z_{s,t}\}_{t=1}^{T_s}$ make up time series $s$ with length $T_s$. The total likelihood for the model with weight $\alpha$ and parameters $\delta, \mathbf{\Gamma}, \theta$ and $\beta$ is thus $\prod_{s=1}^{S} \mathcal{L}_\alpha(\mathbf{y}_s, \mathbf{z}_s ; \delta, \mathbf{\Gamma}, \theta, \beta)$, where $\mathcal{L}_\alpha$ is defined in Eq (12).

## Cross validation for selecting $\alpha$

To determine the optimal value of $\alpha$, we recommend the following cross validation procedure. First, define a set of candidate values for $\alpha \in [0, 1]$. At a minimum, we suggest evaluating $\alpha \in$

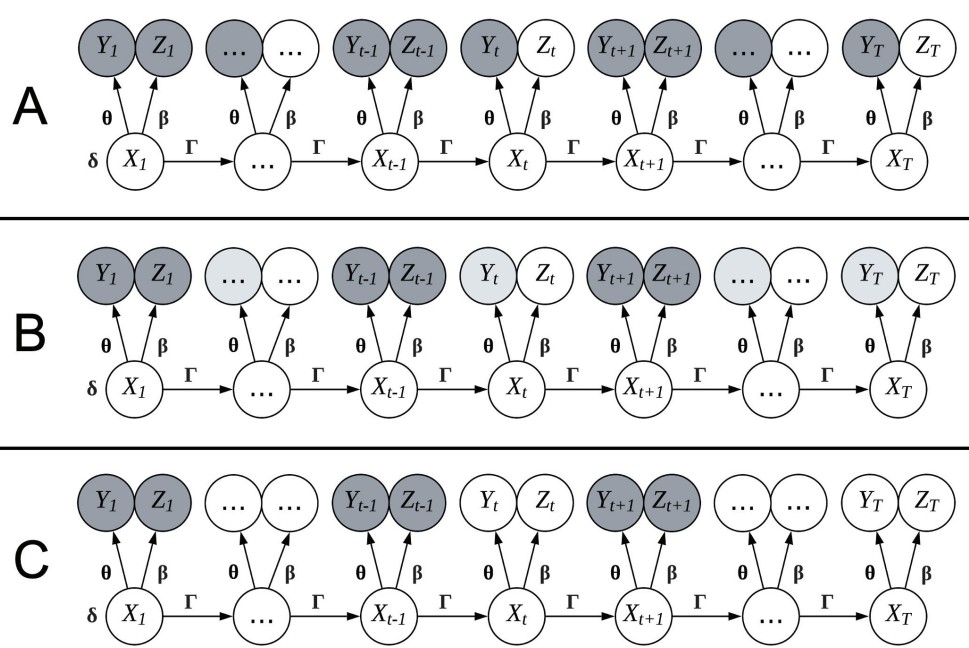

**Fig 2. Graphical representation of PHMMs with different likelihood weightings $\alpha$.** (**a**) PHMM with $\alpha = 1$ that gives equal weight to all observations in the likelihood function. (**b**) PHMM with $\alpha \in (0, 1)$ that down-weights unlabelled observations without ignoring them altogether. (**c**) PHMM with $\alpha = 0$ that ignores all observations that do not have associated label information. The colour (white, light grey, and dark grey) indicates how much a given observation affects the weighted likelihood. White corresponds to treating the random variable as unobserved, dark grey corresponds to treating the variable as fully observed, and light grey corresponds to treating the random variable as observed, but weighting it in the likelihood. The latent states are denoted as $\{X_{t'}\}_{t'=1}^{T}$, the observations are denoted as $\{Y_{t'}\}_{t'=1}^{T}$, and labels are denoted as $\{Z_{t'}\}_{t'=1}^{T}$. In this example, $Z_1$, $Z_{t-1}$, and $Z_{t+1}$ are observed, while all other labels are unobserved (i.e., $Z_{t'} = \varnothing$ for $t' \neq 1, t-1, t+1$).

$\{0, |\mathcal{T}|/(T - |\mathcal{T}|), 1\}$, as the choice $\alpha = |\mathcal{T}|/(T - |\mathcal{T}|)$ approximately balances the contributions of labelled and unlabelled observations in the likelihood function. Although it is technically possible to consider values of $\alpha > 1$, we advise against this, as it assigns greater weight to unlabelled observations relative to labelled ones. Another complication occurs if there exists some hidden state $i \in \{1, \dots, N\}$ with no corresponding labels (i.e., $z_t \neq i$ for all $t \in \{1, \dots, T\}$). In this case we advise against setting $\alpha = 0$ because no labelled data can be used to estimate the state-dependent parameters $\theta^{(i)}$, so the PHMM is unidentifiable.

Next, partition the time series data set into multiple folds for cross validation. If possible, the folds should be the time series themselves $\{\mathbf{y}_s, \mathbf{z}_s\}_{s=1}^S$ to ensure independence between training and test sets. For each fold, train the PHMM using the entire data set minus the selected fold. Apply the forward-backward algorithm to the held-out fold (excluding its associated labels) to estimate the hidden state probabilities $\mathbb{P}(X_{s,t} \mid \mathbf{Y_s} = \mathbf{y_s})$. Repeat this procedure across all folds, yielding cross-validated hidden state probability estimates for the entire data set. The estimated probabilities, together with the true labels, can then be used with standard model validation techniques to assess PHMM performance and determine the most appropriate choice of $\alpha$. Fig 3 displays a diagram of this cross validation process.

Due to its computational complexity, this procedure can be computationally expensive, particularly for large data sets. However, computational efficiency can be improved by training PHMMs on different folds in parallel. In our case studies, we leveraged the Cedar Compute Canada cluster to execute cross validation in parallel, significantly reducing computation time.

## Case studies

We conducted two case studies that use PHMMs to model the behaviour and foraging success of 11 resident killer whales (nine northern, two southern) off the coast of British Columbia, Canada. These case studies were primarily intended to test the predictive performance of the PHMM and demonstrate the process of applying it to ecological data. However, we also performed these case studies for their ecological significance. As mentioned earlier, understanding killer whale foraging and successful prey capture has been a research focus for years, as differences in foraging success may explain why the southern resident killer whale population is doing poorly compared to the northern residents [15,18]. Thus, the results from these case studies can help ecologists correctly predict foraging behaviours that are meaningful for conservation.

The data for these case studies were collected with written approval under the University of British Columbia Animal Care Permit no. A19-0053, Fisheries and Oceans Canada Marine Mammal Scientific License for Whale Research no. XMMS 6 2020, and United States Department of Commerce, NOAA, National Marine Fisheries Service Permit No. 23220. Collection took place in August and September 2020 off the coast of British Columbia in Queen Charlotte Sound, Queen Charlotte Strait, Johnstone Strait, and Juan de Fuca Strait. All resident killer whales were tagged with suction-cup attached Customized Animal Tracking Solutions (www.cats.is) as described by McRae et al. [25]. These tags were deployed using an adjustable 6–8 metre carbon fibre pole and detached using galvanic releases. Post-deployment, the instruments were retrieved utilizing a combination of a Wildlife Computers 363C SPOT tag (providing Argos satellite positions), a goniometer, an ultra high frequency receiver, and a yagi antenna. The tags were equipped with an array of instruments, including 3D kinematic sensors (accelerometer, magnetometer, gyroscope), a time-depth recorder, a 96 kHz HTI hydrophone, and a camera. The time-depth recorder (TDR) and inertial sensors were set to sample at a frequency of 50 Hz. Depth readings were calibrated using a MATLAB package

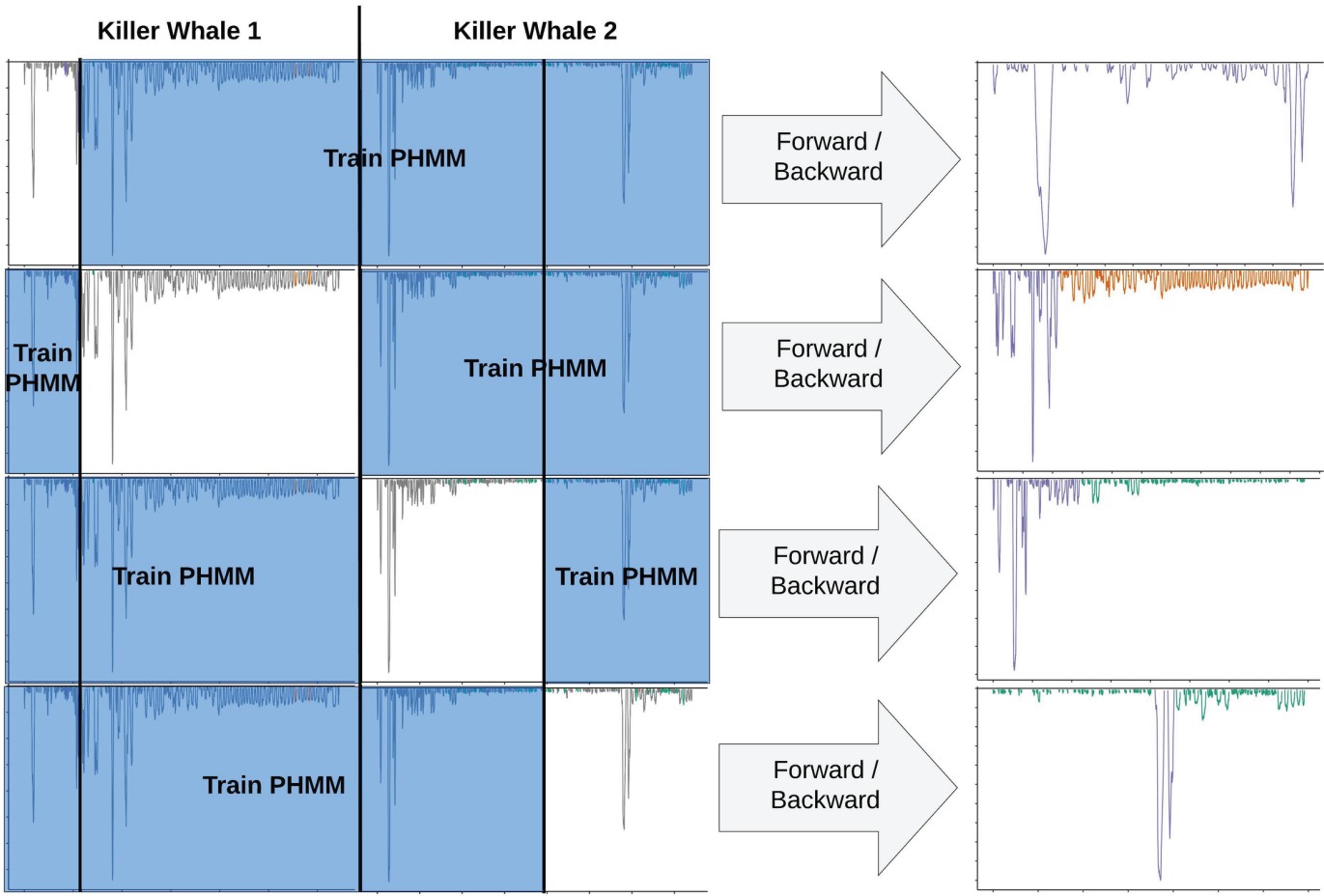

**Fig 3. Cross validation procedure for a PHMM with given $\alpha$.** Dive profiles from two killer whales are divided into four subprofiles, treated as independent time series, and used as folds in cross validation. Each fold is held out, a PHMM is trained on the remaining data set, and the forward/backward algorithm is run on the held out data using the fitted PHMM. The labels of the held out time series are ignored when running the forward-backward algorithm. The estimated labels of the held-out time series are shown in the far right column and used to calculate accuracy measures such as sensitivity, specificity, area under the ROC curve, etc.

developed by Cade et al. [36], which allowed for the extraction of heading, pitch, and roll, as well as three-dimensional dynamic acceleration within the reference frame of the killer whale.

Using these data, we developed two PHMMs to identify killer whale foraging behaviours at different scales. The first PHMM estimated killer whale dive types using individual dives as observations, with some dives labelled as resting, travelling, or foraging using aerial drone recordings of the tagged whales. The second PHMM estimated prey capture events using high-frequency biologging data as observations, with some observations labelled using underwater video and audio recordings of prey capture. For both case studies, we modelled each killer whale as independent, but with shared parameters for their associated PHMMs.

## Case study 1: behavioural classification of killer whale dives

For our first case study, we assigned a latent behaviour to every killer whale dive. To this end, we modelled the data from each killer whale as a sequence of dives and modelled each sequence with a PHMM. The hidden Markov chain was a sequence of dive types and the

observations were summary statistics of each dive. We used three well-known killer whale behaviours as possible dive types: resting, travelling, and foraging [37,38]. We also used aerial and underwater recordings to label a small subset of dives with one of these dive types [25].

**Data processing.** We defined a dive as any period in which the killer whale was below a depth of 0.5 metres for at least 30 seconds, which includes only biologically meaningful dives and excludes surface behaviours. In line with previous studies [7,25], we summarized each dive with its maximum depth and total duration. Formally, the observation associated with dive profile $s$ and dive $t$ was denoted as $y_{s,t} = (m_{s,t}, d_{s,t})$, where $m_{s,t}$ corresponds to maximum depth in metres and $d_{s,t}$ corresponds to dive duration in seconds.

Using drone videos, we visually identified three diving behaviours: resting, travelling, and foraging (classification criteria are given in McRae et al. [25]). Formally, the label associated with whale $s$ and dive $t$ was denoted as $z_{s,t} \in \{\varnothing, 1, 2, 3\}$, where each value of $z_{s,t}$ corresponds to either no label (if $z_{s,t} = \varnothing$), resting (if $z_{s,t} = 1$), travelling (if $z_{s,t} = 2$), or foraging (if $z_{s,t} = 3$). Scatter plots of the dive duration and dive depth of all 11 whales are shown in Appendix S2.

There were a total of 11 killer whales, but the distribution of labels was uneven between individuals (e.g., killer whale A113 had 71 labelled travelling dives, while all other whales combined had 12). To spread the labels between dive profiles more evenly, we randomly divided each killer whale dive profile into two 'subprofiles' so that the two subprofiles had an equal number of labels. Although the two subprofiles are dependent in time, we treated them as independent for computational simplicity. See Fig 3 for an example of splitting two profiles into four subprofiles for cross validation. This process resulted in $S = 22$ killer whale subprofiles, a total of $T = 2169$ dives, and $|\mathcal{T}| = 106$ labels of dive types.

**Model formulation.** We used a PHMM with $N = 3$ dive types to match the drone-identified labels of resting, travelling, and foraging dives. For whale $s$ and dive $t$, we denoted the hidden dive type as $X_{s,t} \in \{1, 2, 3\}$. Histograms and scatter plots revealed that dive duration and maximum depth looked to be distributed approximately as mixtures of log-normal distributions, and the two features were also highly correlated (see Appendix S2 for scatter plot). Therefore, we set the joint, state-dependent distribution of the observations to be a bivariate log-normal distribution.

Recall that we used thresholds of $m_{s,t} \geq 0.5$ and $d_{s,t} \geq 30$ to define biologically relevant dives. However, we modelled these observations using a two-dimensional log-normal distribution whose sample space is $\mathbb{R}^2_{>0}$, so our model is misspecified. We could use a truncated multivariate log-normal distribution instead, but many HMM software packages do not incorporate truncated log-normal distributions by default [39,40], and several ecological studies make this modelling choice as well [7,11,41]. We therefore use a non-truncated log-normal distribution for simplicity and reproducibility. Labels were identified with high confidence, so we defined $g^{(i)}$ according to Eq (7).

**Model evaluation** We fit five different candidate PHMMs corresponding to five different values of $\alpha$. We tested $\alpha = 0$, which ignores all unlabelled data; $\alpha = 1$, which gives equal weight to all observations; and $\alpha = |\mathcal{T}|/(T - |\mathcal{T}|) = 0.049$, which approximately balances the contribution of labelled and unlabelled observations. For completeness, we also tested $\alpha = 0.025$, which averages $\alpha = 0$ and $\alpha = 0.049$; and $\alpha = 0.525$, which averages $\alpha = 0.049$ and $\alpha = 1$. All PHMMs, including those within the cross validation procedure, were fit using 10 random restarts and a custom version of the momentuHMM package in R [39,42].

We used the 22 subprofiles as folds in cross validation to estimate the probability of each dive's type conditioned on the observations (i.e., $\mathbb{P}(X_{s,t} = i \mid \mathbf{Y}_s = \mathbf{y}_s)$ for $s = 1, \ldots, 22$; $t = 1, \ldots, T_s$; and $i = 1, 2, 3$). See Fig 3 for a diagram of this cross validation procedure. Next, we calculated the sensitivity, specificity, and area under the receiver operating characteristic curve (AUC) associated with each dive type. We calculated sensitivity for dive type $i$ as the

average estimated probability that a dive has type $i$, averaged across all dives that were confirmed via drone to have type $i$. Likewise, we calculated specificity for dive type $i$ as the average estimated probability that a dive does not have type $i$, averaged across all dives that were confirmed via drone to *not* have type $i$. AUC balances sensitivity and specificity and takes values between 0 and 1, where higher is better [43]. We also ran the Viterbi algorithm on each fold within the cross validation procedure and plotted the resulting dive profiles as a visual model evaluation tool [44].

As a baseline, we implemented three machine learning methods to predict dive types: multinomial logistic regression using the `nnet` package in R (MLR), random forests using the `randomForest` package in R (RF), and support vector machines with a radial kernel using the `e1071` package in R (SVM) [45–47]. Each model was trained using default package settings, and performance was evaluated using the cross validation procedure described earlier to estimate sensitivity, specificity, and AUC. These fully-supervised, single-frame methods used only labelled data and did not account for temporal structure. While we report the accuracy of these baselines as a comparison, PHMMs offer several key advantages. Most notably, PHMMs model both the hidden states and the underlying generative temporal process, unlike the baseline methods which focus solely on prediction.

**Results.** Except for when $\alpha = 1$, the PHMMs are either better than or comparable to the single frame methods across all criteria and behaviours (see Fig 4). The random forest and multinomial logistic regression models had worse AUC values for all behaviours compared to all PHMMs. The SVM model's AUCs for resting (0.89) and travelling (0.91) were comparable to the PHMM with $\alpha = 0.05$ (0.87 and 0.93, respectively), but the SVM model's AUC for foraging (0.38) was much worse than all of the PHMMs (>0.9 for all $\alpha$). This result is especially striking because foraging is a behaviour of particular ecological significance.

The PHMMs with $\alpha$ strictly between 0 and 1 tended to obtain better results for cross-validated sensitivity, specificity, and AUC, demonstrating the effectiveness of the weighted likelihood approach (Fig 4). Compared to the PHMMs with $\alpha = 0$ and $\alpha = 1$, the PHMM with $\alpha = 0.049$ had the best sensitivity for foraging dives and travelling dives, and it had the second-best sensitivity for resting dives. It had the best specificity for resting dives and travelling dives, and its specificity for foraging dives was comparable to the other models. In addition, the PHMM with $\alpha = 0.049$ had an AUC for resting that was comparable or better than the other PHMMs (0.866), but it had the best AUC for foraging (0.955) and the best AUC for travelling (0.926). Results for the PHMMs with $\alpha = 0.025$ and $\alpha = 0.525$ are similar to the PHMM with $\alpha = 0.049$ (see Appendix S1).

The PHMMs with $\alpha \in (0, 1)$ were also more biologically interpretable compared to those with $\alpha = 0$ and $\alpha = 1$. Namely, for the PHMMs with $\alpha \in (0, 1)$, the time series of decoded dive types contained long sequences of a single dive type (Fig 5). This behaviour matches prior studies that model killer whale behaviours as lasting from tens of minutes to hours [25]. In comparison, the PHMM with $\alpha = 1$ resulted in a dive profile that switched relatively frequently between foraging and resting dives (e.g., Fig 5D). The PHMM with $\alpha = 0$ produced better results than the PHMM with $\alpha = 1$, but it still estimated some rapidly switching dive types (e.g., Fig 5B). Appendix S2 also displays scatter plots of maximum depth and dive duration data from all whales, coloured by cross-validated, Viterbi-decoded dive type.

The PHMM that completely ignored unlabelled dives ($\alpha = 0$) obtained better results than the PHMM that fully included the unlabelled dives ($\alpha = 1$). While one may expect that more data should improve accuracy, including unlabelled data is known to degrade the performance of a classifier in certain situations [48]. It is thus natural that the optimal approach for this case study neither fully included nor totally ignored the labelled dive types.

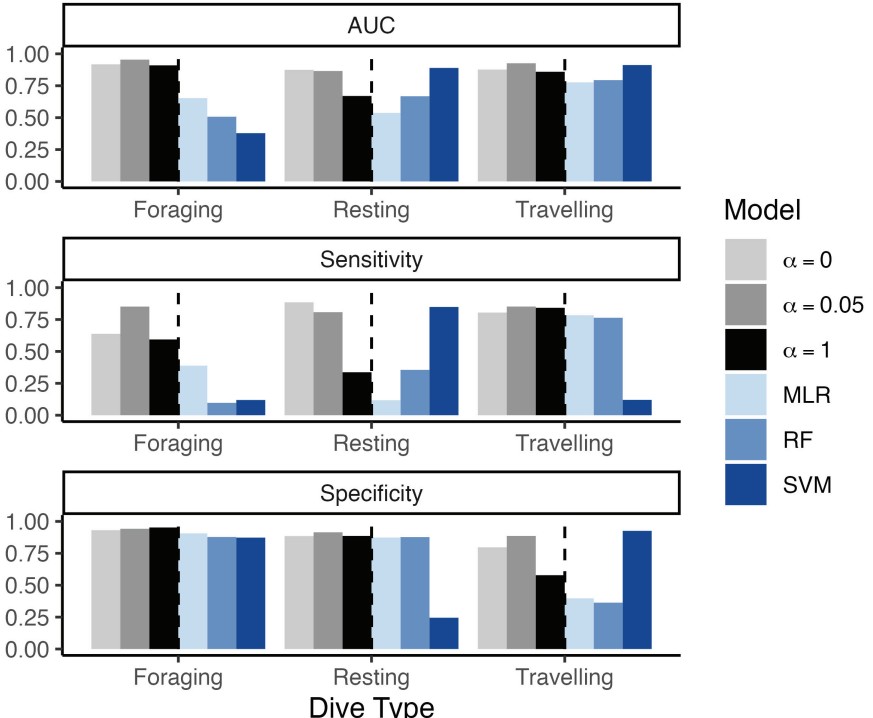

**Fig 4. Sensitivity, specificity, and AUC values associated with each dive type.** True values are determined via the drone-detected dive types. All models

Finally, we used the PHMM with $\alpha$ = 0.049 to estimate how often the northern and southern resident killer whales engaged in foraging behaviour. In particular, we fit the PHMM to the entire data set, including all labels, and then ran the forward-backward algorithm on all subprofiles and dives to calculate $\mathbb{P}(X_{s,t} = 3 \mid \mathbf{Y}_s)$ for $t = 1, \dots, T_s$ and $s = 1, \dots, 22$. We then labelled dive $t$ of whale $s$ as foraging if its decoded probability of foraging was above 50%. Using this procedure, we estimated that southern resident killer whales foraged for 5.47 hours, or 32.0% of the time, and that northern residents foraged for 17.90 hours, or 26.8% of the time. This sample size was very small (2 southern residents and 9 northern residents), and the sample of southern residents in particular was made up entirely of adult males killer whales, which are expected to forage more than other age-sex groups [15]. Nonetheless, our results are consistent with Tennessen et al. [15], who found that southern residents spent less time travelling and resting compared to northern resident killer whales.

## Case study 2: identification of killer whale foraging

The second case study focused on identifying successful foraging events within the killer whale dives. Therefore, we divided all dives deeper than 30 metres into sequences of two-second windows and modelled each sequence with a PHMM. The hidden Markov chain was a sequence of unobserved two-second subdive states, and the observations were summary statistics calculated from each two-second window. We defined six possible subdive state behaviours: descent, bottom, prey chase, prey capture, ascent with a fish, and ascent without a fish.

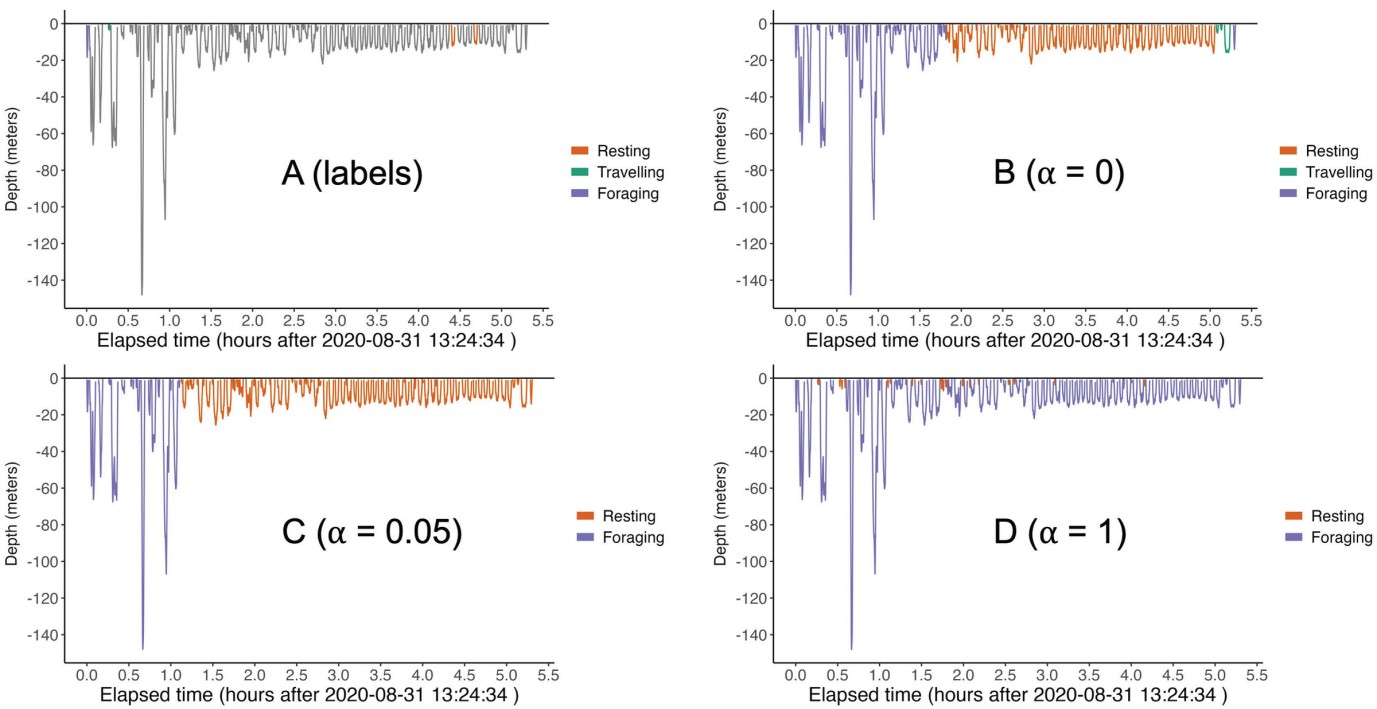

**Fig 5. Viterbi-decoded dives of killer whale D26 (male, 10 years old) using different PHMMs.** Each PHMM was fit to the data set with the subprofile held out. Then the Viterbi algorithm was used on the held-out data set with its labels removed in order to test the predictive performance of each PHMM.

**Data processing.**  We defined a killer whale dive identically to the previous case study, but only included dives deeper than 30 metres because Wright et al. [37] found that killer whale prey captures usually occur below that depth. Adult Chinook have also been found throughout the water column to depths exceeding 100 metres, particularly in areas where the risk of predation appears high [49,50]. Dives shallower than 30 metres likely follow a substantially different distribution than foraging dives deeper than 30 metres, and we solely focus on modelling the deeper foraging dives in this case study.

We divided each dive into two-second windows and calculated summary statistics that were identified by Tennessen et al. [51] to be indicative of foraging behaviour: change in depth ($d_{s,t}$ for dive $s$ and window $t$), heading total variation ($h_{s,t}$), and jerk peak ($j_{s,t}$). Change in depth was defined as the last depth reading of the window minus the first depth reading of the window in metres. Heading total variation was determined by calculating the difference between heading readings every 1/50 of a second (in radians), taking the absolute value of that difference, and summing up all of the differences over the course of the two-second window. Jerk peak was calculated by taking the difference between acceleration vectors every 1/50 of a second (in metres per second squared), taking the magnitude of that difference, and then calculating the maximum over each two-second interval. To adjust for variation between dives and tags, we also divided the jerk peak of each two-second window by the median jerk peak for the bottom 70% of its corresponding dive [51]. In summary, the observation associated with dive $s$ and window $t$ was denoted as $y_{s,t} = (d_{s,t}, h_{s,t}, j_{s,t})$. This process resulted in a total of $S = 130$ dives and $T = 15821$ two-second windows.

To label windows associated with prey capture, we used a process inspired by previous work on killer whale foraging [37,51]. First, crunching sounds associated with prey handling

were identified from the hydrophone using the behavioural analysis software BORIS [52]. Crunching sounds associated with foraging were corroborated using video evidence of prey handling as well as audio observations of echolocation clicks. Next, Wright et al. [37] found that killer whales catch prey immediately before ascending, so we ignored crunches that occurred more than 30 seconds before a dive's 'ascent' phase as defined by Tennessen et al. [51]. Namely, the 'ascent' phase began the moment after the killer whale achieved a depth at least 70% of its maximum dive depth. If the first non-ignored crunch occurred *before* 'ascent', we labelled its window as 'prey capture'. If the first non-ignored crunch was heard *during* 'ascent', then the exact moment of prey capture was ambiguous, so we labelled the final window of the dive as 'ascent with a fish'.

We were also able to obtain some window labels not associated with prey capture. First, if the video recorder was on for the entire dive, but there was no audible crunch or visual indication of foraging (e.g., scales), then we labelled the final window of the dive as 'ascent without a fish'. Second, we labelled the first window of every dive as 'descent'. Formally, the label associated with window $t$ of dive $s$ was denoted as $z_{s,t} \in \{\varnothing, 1, 2, 3, 4, 5, 6\}$, where each possible value of $z_{s,t}$ corresponds to no label ($z_{s,t} = \varnothing$), descent ($z_{s,t} = 1$), bottom ($z_{s,t} = 2$), chase ($z_{s,t} = 3$), capture ($z_{s,t} = 4$), ascent without a fish ($z_{s,t} = 5$), or ascent with a fish ($z_{s,t} = 6$). In total, we labelled 130 windows as 'descent, five windows as "capture", two windows as 'ascent with a fish', and 19 windows as 'ascent without a fish'. This resulted in $|\mathcal{T}| = 156$ labels, which make up less than 1% of the $T = 15821$ windows in total. Scatter plots showing the observed data for all two-second windows are shown in Appendix S2.

**Model formulation.** As mentioned before, we defined a PHMM with $N = 6$ subdive states: descent, bottom, chase, capture, ascent without a fish, and ascent with a fish. We assumed that all dives shared an initial distribution $\delta = \begin{pmatrix} 1 & 0 & 0 & 0 & 0 & 0 \end{pmatrix}$ and transition probability matrix $\Gamma$ as shown in Eq (13).

$$\Gamma = \begin{array}{c} \\ \\ \\ \\ \\ \\ \end{array} \overset{\begin{array}{cccccc} \text{descent} & \text{bottom} & \text{chase} & \text{capture} & \text{ascent w/o fish} & \text{ascent w/ fish} \end{array}}{\begin{pmatrix} \Gamma^{(1,1)} & \Gamma^{(1,2)} & 0 & 0 & \Gamma^{(1,5)} & 0 \\ 0 & \Gamma^{(2,2)} & \Gamma^{(2,3)} & 0 & \Gamma^{(1,5)} & 0 \\ 0 & \Gamma^{(3,2)} & \Gamma^{(3,3)} & \Gamma^{(3,4)} & \Gamma^{(3,5)} & 0 \\ 0 & 0 & 0 & \Gamma^{(4,4)} & 0 & \Gamma^{(4,6)} \\ 0 & 0 & 0 & 0 & 1 & 0 \\ 0 & 0 & 0 & 0 & 0 & 1 \end{pmatrix}} \begin{array}{l} \text{descent} \\ \text{bottom} \\ \text{chase} \\ \text{capture} \\ \text{asc w/o fish} \\ \text{asc w/ fish} \end{array}$$

(13)

For an intuitive visualization of how the Markov chain can evolve, see Fig 6. In short, this transition probability matrix reflects that marine mammal dives have distinct descent, bottom, and ascent phases [51], and that killer whales begin their ascent phase immediately after prey capture [37]. We also divided the bottom phase to include a low activity state (bottom) and a high activity state (chase), which is in line with results from Sidrow et al. [53].

Given that $X_{s,t} = i$, we modelled change in depth with a normal distribution, heading total variation with a gamma distribution, and normalized jerk peak with a gamma distribution. To enforce that the killer whale does not ascend or descend on average during the bottom of its dive, we set the state-dependent distributions of change in depth for the bottom, chase, and capture states to have a mean of zero. Further, we wanted to ensure that the model distinguished foraging based primarily on the 'prey capture' subdive state rather than differences between the 'ascent with a fish' and 'ascent without a fish' states. As such, we set the two state-dependent distributions associated with ascent to be identical. Given the hidden state of a

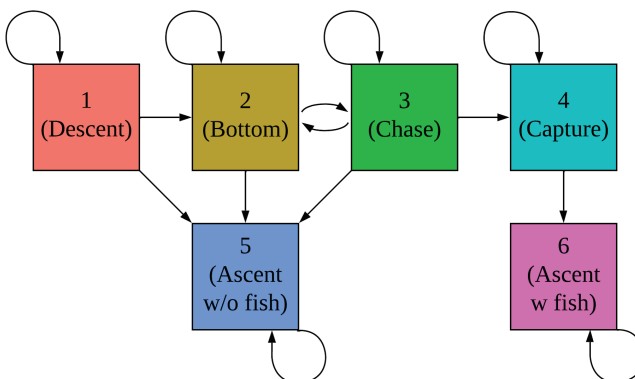

**Fig 6. Visualization of how the Markov chain $X_s$ evolves for killer whale foraging dives.** Arrows correspond to non-zero entries in $\boldsymbol{\Gamma}$, where arrows point from row number to column number. Within a dive, a killer whale can proceed from descent, to bottom, to chase, to capture, to ascent with fish. It can also ascend without a fish at any time before capture, and it can switch between the bottom and chase states freely.

window $(X_{s,t})$, we also assumed that all summary statistics were independent. Subdive state labels were identified with high confidence, so we defined $g^{(i)}$ according to Eq (7).

**Model evaluation.** We fit five different PHMMs corresponding to $\alpha \in \{0.0001, 0.001, 0.01, 0.1, 1\}$ using 10 random restarts and a custom version of the momentuHMM package in R [39,42]. We did not include $\alpha = 0$ in this case study because we did not have labels associated with subdive states 2 and 3 (bottom and chase). Thus, if $\alpha = 0$, there were no observations that could be used to estimate the parameters of the state-dependent distribution parameters $\theta^{(2)}$ and $\theta^{(3)}$, and the model would not be identifiable.

In addition to the PHMMs, we also implemented several single-frame baseline methods to identify prey capture using dive-level summary statistics. First, Tennessen et al. calculated three summary statistics for each dive: (1) the maximum of jerk during the bottom 70% of the dive, divided by the median jerk during the same period, (2) the absolute value of the killer whale's roll at the moment of jerk peak, and (3) the circular variance of heading during the bottom 70% of the dive. Then, Tennessen et al. took the minimum value of every summary statistic over all confirmed prey capture dives to obtain thresholds for each of the three dive-level summary statistics. Finally, a dive was labelled as a successful foraging dive if *every* dive-level summary statistic surpassed that threshold. Using these three dive-level summary statistics, we also implemented Firth's bias-reduced logistic regression [54,55], random forests [46], and support vector machines using a radial kernel [47]. Appendix S2 shows scatter plots of normalized jerk peak, average bottom heading total variation, and roll at jerk peak for all dives deeper than 30 metres.

We evaluated each model based on how well it predicted successful foraging dives. Note that the probability that dive *s* is a successful foraging dive is equal to the probability that it ends in either the 'capture' or 'ascent with a fish' state. In other words, it is the probability that window $T_s$ of dive *s* has subdive state 4 (capture) or subdive state 6 (ascent with a fish), i.e. $\mathbb{P}(X_{s,T_s} \in \{4,6\} \mid \mathbf{Y}_s = \mathbf{y}_s)$. To estimate this value, we randomly and equally split the seven labelled successful foraging dives and 19 labelled dives without successful foraging into four folds and performed cross validation with the forward-backward algorithm. We calculated AUC values within each fold and reported their averages. Finally, we also fit each PHMM to the entire data set to visualize their state-dependent distributions.

**Results.** The PHMM with $\alpha = 0.0001$ had an average AUC of 0.90, the PHMM with $\alpha = 1.0$ had an AUC of 0.96, and the PHMM with $\alpha = 0.01$ had a perfect AUC of 1 across all four folds (Fig 7). Thus, using $\alpha \in (0, 1)$ (namely $\alpha = 0.01$ here) improved predictive performance over $\alpha = 0$ and $\alpha = 1$.

Our method outperformed the baseline of Tennessen et al. [51], which had an average AUC of 0.79. The PHMM with $\alpha = 0.01$ had the same AUC value as two single-frame methods (logistic regression and support vector machine), but the PHMM method has the additional benefit of labelling dive phases in addition to determining successful foraging dives. One possible reason that several methods achieve a perfect AUC score is that this data set contains only 7 positive examples of successful foraging. Thus, determining a granular estimate of accuracy is difficult.

The biological interpretation of each PHMM heavily depended on the weight $\alpha$. For example, the 'bottom' and 'chase' states looked very similar for the PHMMs with $\alpha \leq 0.01$, but the two states were better separated within the PHMM with $\alpha = 1$ (Fig 8). However, the 'bottom' subdive state had higher mean jerk peak and heading total variation than the 'chase' subdive state for the PHMM with $\alpha = 1$. These results are the opposite of what ecologists expect biologically, indicating that a large separation between state-dependent distributions is not always more biologically interpretable. We conjecture that the 'bottom' and 'chase' subdive states are particularly unintuitive partially because there are no labels associated with either of these subdive states (i.e. $z_{s,t} \neq 2, 3$ for any $s$ or $t$). As a result, there is little information for the model to differentiate the two subdive states.

Differences between each model's distribution for the capture state appeared to impact predictive performance. For example, the PHMM with $\alpha = 0.0001$ estimated a relatively high mean and low standard deviation for heading total variation (middle panel of Fig 8a). As such, it failed to identify a prey capture event in which heading total variance was highly variable (Fig 9). Alternatively, the PHMM with $\alpha = 1$ estimated a relatively low mean for heading variation (middle panel of Fig 8c). As a result, it was often too sensitive and decoded some dives that lacked successful foraging as containing a prey capture state (Fig 10). The PHMM with $\alpha = 0.01$ estimated a high mean and a high standard deviation for heading total variation (middle panel of Fig 8b). It correctly identified a wide variety of labelled foraging dives,

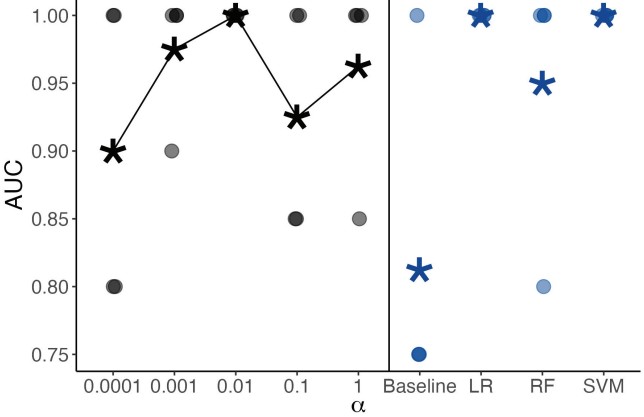

**Fig 7. Four-fold, cross-validated area under the ROC curve values for PHMMs and baseline methods**. Horizontal jitters have been added because dots occasionally fall on top of one another. Averages are shown as stars and connected with a line for the PHMM approaches. The 'Baseline' approach refers to Tennessen et al. [51].

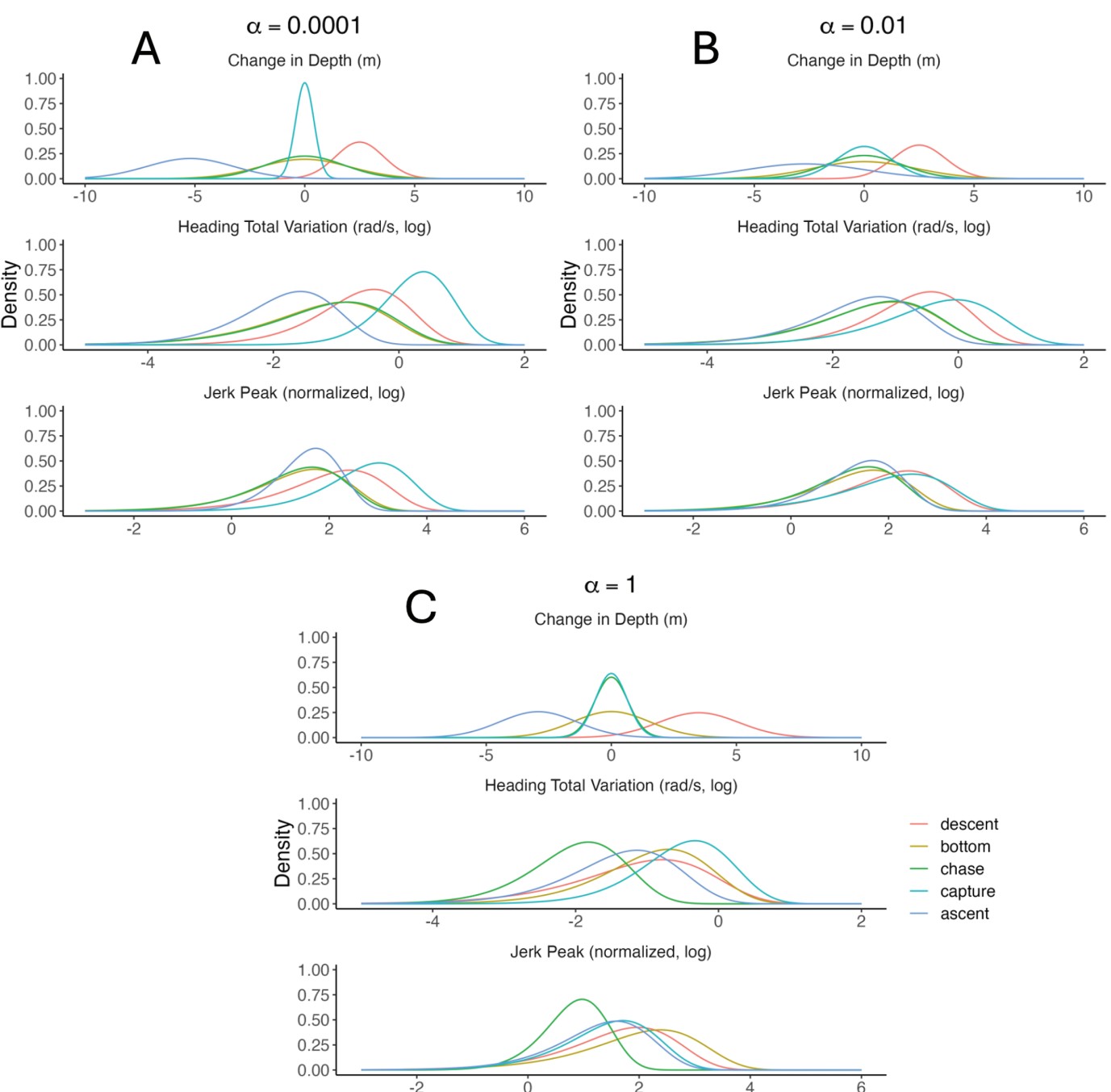

**Fig 8. State-dependent densities of each observation for PHMMs with different values of $\alpha$.** Observations include change in depth (top panels), heading total variation (middle panels), and normalized jerk peak (bottom panels) for PHMMs with $\alpha$ = 0.0001 (top left), $\alpha$ = 0.01 (top right), and $\alpha$ = 1 (bottom). Densities are coloured according to their corresponding subdive state. Parameters were estimated using the entire killer whale data set (i.e. no cross validation was performed). For a given observation $Y_{s,t}$, all features were assumed to be independent after conditioning on the subdive state $X_{s,t}$. Note that the 'ascent with a fish' and 'ascent without a fish' states were assumed to have identical distributions, so both are listed simply as 'ascent'.

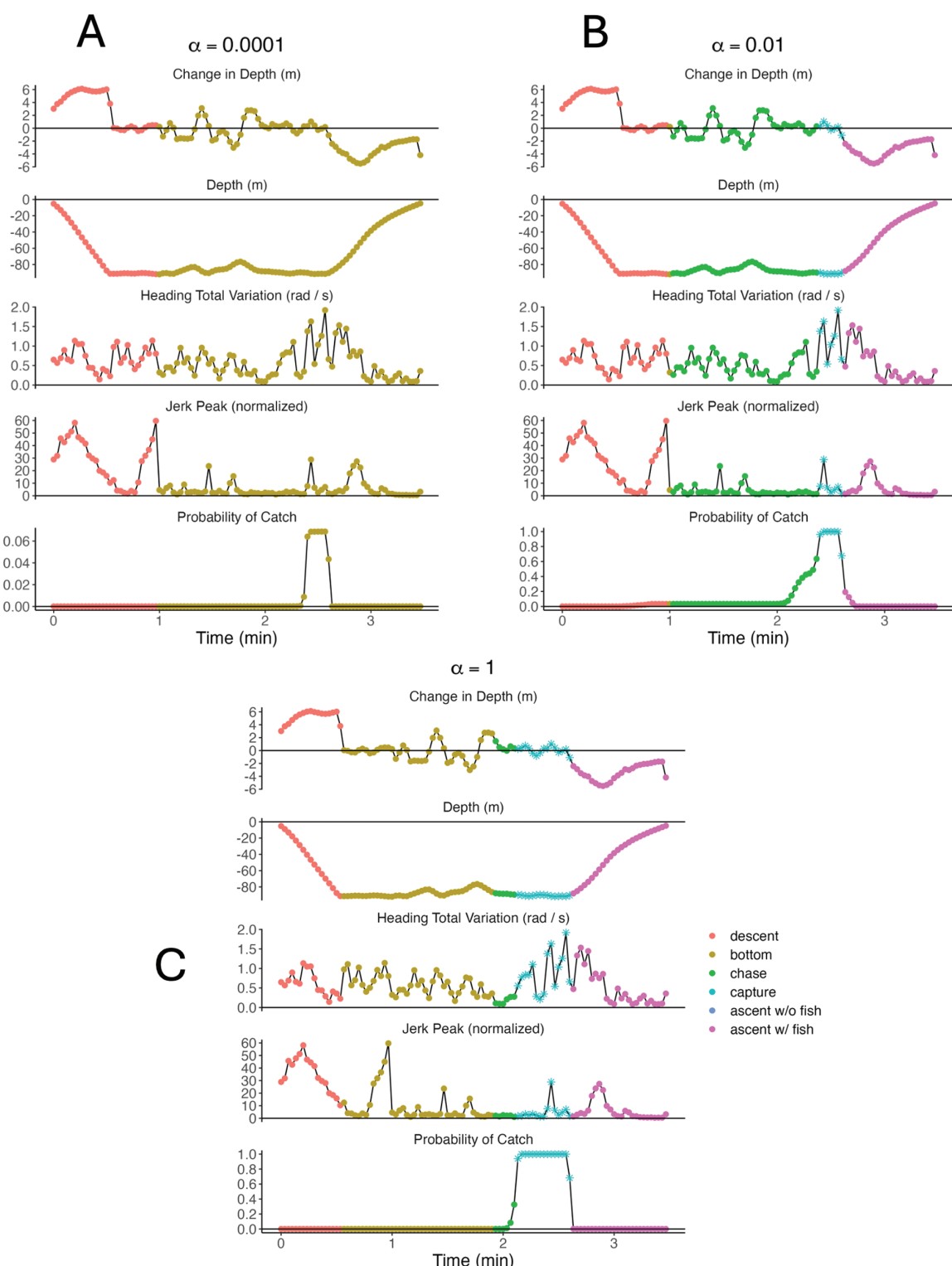

**Fig 9. Dive profiles, observations, and decoded hidden state probabilities for selected successful foraging dive.** PHMMs with $\alpha = 0.0001$ (top left), $\alpha = 0.01$ (top right) and $\alpha = 1$ (bottom) are shown. Each subplot displays change in depth (top panel), raw depth (second panel), heading total variation (third panel), normalized jerk peak (fourth panel), and probability of 'capture' (bottom panel). Observations are coloured according to the most-likely sequence of hidden states as determined by the Viterbi algorithm. The estimated probability of successful foraging, $\mathbb{P}(X_{s,T_s} \in \{4, 6\} \mid \mathbf{Y}_s = \mathbf{y}_s)$, is 0.069 for $\alpha = 0.0001$, 1 for $\alpha = 0.01$, and 1 for $\alpha = 1$.

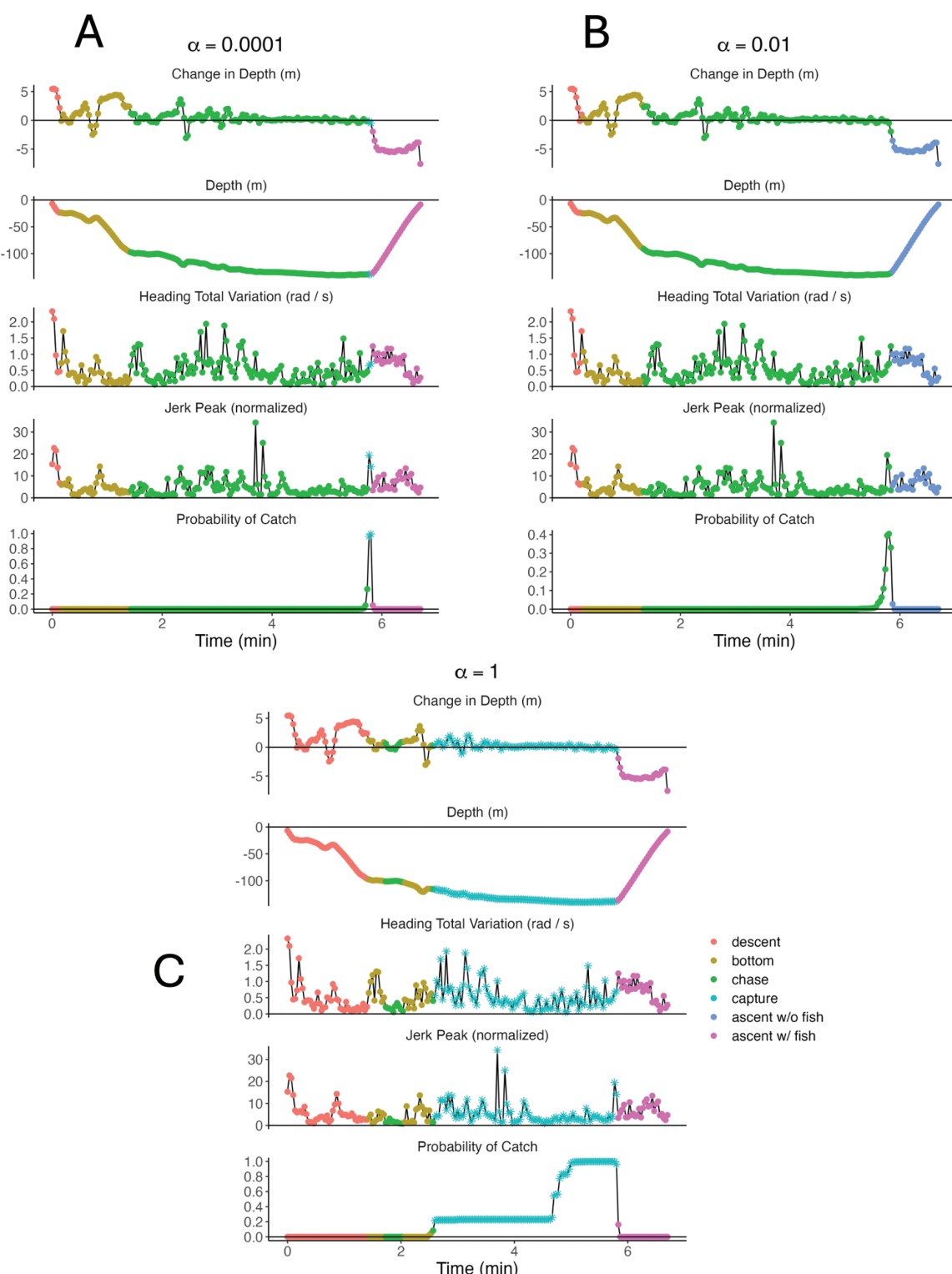

**Fig 10. Dive profiles, observations, and decoded hidden state probabilities for selected dive without foraging.** PHMMs with $\alpha = 0.0001$ (top left), $\alpha = 0.01$ (top right) and $\alpha = 1$ (bottom) are shown. Each subplot displays change in depth (top panel), raw depth (second panel), heading total variation (third panel), normalized jerk peak (fourth panel), and probability of 'capture' (bottom panel). Observations are coloured according to the most-likely sequence of hidden states as determined by the Viterbi algorithm. The estimated probability of successful foraging, $\mathbb{P}(X_{s,T_s} \in \{4, 6\} \mid \mathbf{Y}_s = \mathbf{y}_s)$, is 1 for $\alpha = 0.0001$, 0.414 for $\alpha = 0.01$, and 1 for $\alpha = 1$.

but it was not overly sensitive and correctly identified low-activity dives as lacking successful foraging.

Finally, we used the PHMM with $\alpha = 0.01$ to estimate the total number of successful foraging dives from southern and northern resident killer whales in this data set. In particular, we fit the full model to the entire data set, including labels, and then ran the forward-backward algorithm on all unlabelled dives to estimate the probability that each unlabelled dive was a successful foraging dive, $\mathbb{P}(X_{s,T_s} \in \{4,6\} \mid \mathbf{Y}_s = \mathbf{y}_s)$ for $s = 1, \ldots, 130$. Then, we labelled dive $s$ as a successful foraging dive if the estimate of this probability was above 50%. This process resulted in 6 estimated successful foraging dives from southern resident killer whales and 37 estimated successful foraging dives from northern resident killer whales. After combining these results with those from the first case study, we found that southern resident killer whales caught an average of 1.03 fish per hour of foraging effort, while northern resident killer whales caught an average of 2.00 fish per hour of foraging effort. These results support the finding that northern resident killer whales have more foraging success compared to southern residents [15]. However, our sample size is small (2 southern resident killer whales and 9 northern resident killer whales), and the tag attachments were relatively short and thus provided only a snapshot in time. For example, some killer whales were tagged right after foraging, so a tag attachment of several hours recorded no foraging events. Future studies can use the methods outlined here with larger sample sizes and longer tag attachments to further investigate the differences between northern and resident killer whale foraging success.

## Simulation study

We undertook a simulation study to investigate how different data-generating conditions influence the accuracy and optimal choice of $\alpha$ within a PHMM. These simulations allow us to systematically assess the impact of key parameters on model performance and guide the selection of $\alpha$ in practical applications.

We ran a total of 11 experiments. Each experiment consisted of simulating 100 data sets, fitting PHMMs with multiple values of $\alpha$ to those data sets, and evaluating the performance of the PHMMs. We first describe the process used to generate synthetic data in a series of controlled experiments. Then, we describe how we fit the PHMMs to each simulated data set and evaluated their accuracy.

### Data simulation

As a control experiment, we simulated 100 data sets drawn from an HMM to obtain hidden states $\mathbf{X} = \{X_t\}_{t=1}^{T}$ and observations $\mathbf{Y} = \{Y_t\}_{t=1}^{T}$. Each data set was a single time series (i.e., $S = 1$) made up of $T = 2000$ observations from a PHMM with $N = 2$ hidden states. The probability transition matrix was set to

$$\Gamma = \begin{pmatrix} 0.99 & 0.01 \\ 1-\gamma & \gamma \end{pmatrix}, \tag{14}$$

and the initial distribution was set to the stationary distribution of $\Gamma$, $\delta = \begin{pmatrix} (1-\gamma)/(1.01-\gamma) & 0.01/(1.01-\gamma) \end{pmatrix}$. For the control experiment we set $\gamma = 0.95$. The state-dependent distributions were shifted $t$-distributions with $\nu = 4$ degrees of freedom, implying a standard deviation of $\sqrt{\nu/(\nu-2)}$. We separated the two state-dependent distributions by $\kappa = 1$ standard deviation, so

$$Y_t = \begin{cases} -\frac{\kappa}{2}\sqrt{\frac{\nu}{\nu-2}} + \epsilon_t, & X_t = 1 \\ \frac{\kappa}{2}\sqrt{\frac{\nu}{\nu-2}} + \epsilon_t, & X_t = 2 \end{cases}, \qquad \epsilon_t \sim t(\nu). \tag{15}$$

After simulating **X** and **Y**, we constructed the subset of labelled time indices $\mathcal{T}$ by randomly selecting a proportion of $\ell = 0.01$ of all time indices $t$. The time indices were selected such that at least two labels were selected from both hidden states. Then, we set $Z_t = X_t$ for all $t \in \mathcal{T}$ and $Z_t = \varnothing$ for all $t \notin \mathcal{T}$. Appendix S2 displays a time series generated using this process.

For each of the other 10 experiments, we carried out the same data generation procedure as the control experiment, but we picked exactly one of five settings ($T$, $\gamma$, $\nu$, $\ell$, or $\kappa$) and changed it to be either larger or smaller than the control. Namely, we altered the length of the time series from $T = 2000$ to $T \in \{200, 20000\}$; the bottom row of the transition matrix in Eq (14) from $\gamma = 0.95$ to $\gamma \in \{0.75, 0.99\}$; the degrees of freedom from $\nu = 4$ to $\nu \in \{2.5, 100\}$; the density of labels from $\ell = 0.01$ to $\ell \in \{0.001, 0.1\}$; and the separation between state-dependent distributions from $\kappa = 1$ to $\kappa \in \{0.5, 2\}$.

## Model fitting and evaluation

We modelled the simulated data sets using PHMMs with normal state-dependent distributions. This deliberately introduced model misspecification as the simulated observations were actually drawn from PHMMs with $t$-distributed state-dependent distributions. Each fitted PHMM had $N = 2$ hidden states, which matched the generating process.

For each of the 100 replicated data sets in each of the 11 experimental conditions, we fit six PHMMs corresponding to different weights $\alpha \in \{0, 0.001, 0.01, 0.1, 0.5, 1\}$. For evaluation, we also generated a separate test set using the same generative mechanism as the training set. We then ignored the test set's latent state labels **Z** and applied the forward-backward algorithm using the estimated parameters from each of the six PHMMs. This yielded the estimated hidden state probabilities for all observations in the test set for each of the six PHMMs.

Finally, we evaluated each PHMM's classification performance by comparing its estimated hidden-state probabilities to the true hidden states from the test set. Specifically, we computed the AUC for each combination of experiment, data set, and PHMM using the same calculation from the first case study.

## Results

Fig 11 shows boxplots of the 100 AUC values from each experiment and each value of $\alpha$. For the control experiment (middle column), the value of $\alpha$ that maximizes the median AUC is $\alpha \approx 0.01 = \ell$. As the degrees of freedom $\nu$ grows large (e.g., $\nu = 100$), the optimal value of $\alpha$ approaches 1, indicating that it is more useful to ignore unlabelled data when the model is misspecified. As the separation of the state-dependent distributions increases, the AUCs of all PHMMs increase, but the AUCs for the PHMMs with $\alpha \geq 0.5$ improve the most drastically. This indicates that it is more useful to down-weight unlabelled data when the hidden states are less separated. Next, as $\gamma$ increases from $\gamma = 0.75$ to $\gamma = 0.99$, the classes become more balanced and the optimal value of $\alpha$ increases, indicating that down-weighting unlabelled data is useful when there is more class imbalance in the data set. In addition, as the proportion of labels $\ell$ increases, the optimal value of $\alpha$ also increases, but when $\ell = 0.1$, all PHMMs perform very well. This result is consistent with our conjecture that $\alpha \approx \ell$ approximately balances the contribution of labelled and unlabelled data in the likelihood. Finally, as $T$ increases, the median AUC increases for all PHMMs except for those with $\alpha = 0.5$ and $\alpha = 1$. This implies that even as the amount of data grows large, model misspecification can still lead to erroneous hidden state estimates when using an HMM that equally weights labelled and unlabelled data.

Although we explore ways in which model misspecification can affect the optimal value of $\alpha$ in a PHMM, our experiments are by no means exhaustive (see, for example, [56]). Since

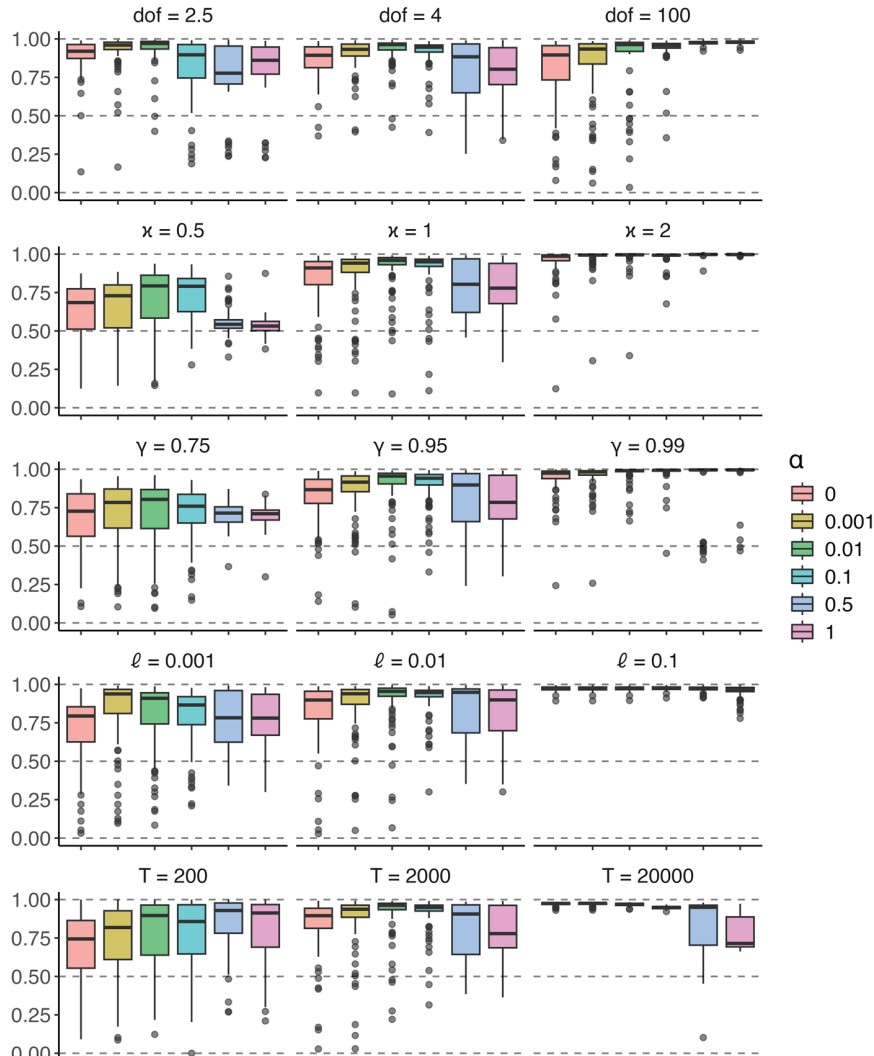

**Fig 11. AUCs of mispecified PHMMs.** The middle column gives results from five runs of the control experiment. Namely, there are $\nu = 4$ degrees of freedom, the separation between state-dependent distributions is $\varkappa = 1$ standard deviations, the bottom-right entry of the transition matrix $\Gamma$ from Eq (14) is $\gamma = 0.95$, the proportion of labelled data is $\ell = 0.01$, and the length of the time series is $T = 2000$. The left and right columns give results from the other ten experiments, each corresponding to changing the value of one setting as indicated by the subplot titles. We denote the degrees of freedom $\nu$ as 'dof' in the figure. Each experiment was repeated on 100 independently generated time series, and the AUCs for all 100 test sets are plotted with boxplots.

models on real data sets are always mispecified, we recommend that practitioners perform cross validation to determine the optimal value of $\alpha$.

## Discussion

In this work, we incorporated sparse labels into an HMM in a natural way that changes the influence of unlabelled observations and demonstrably improves predictive performance. In particular, we weighted observations without associated labels within the likelihood of the HMM using a parameter $\alpha \in [0, 1]$. On the extremes, $\alpha = 0$ corresponds to an HMM which

totally ignores unlabelled data and $\alpha = 1$ corresponds to a traditional, unweighted HMM. We used cross-validated accuracy metrics to find an optimal value of $\alpha$ between these extremes.

We also conducted a simulation study to investigate how the observed data set affects the optimal weight $\alpha$. In particular, down-weighting unlabelled data is beneficial under challenging modelling conditions such as model misspecification, poor hidden state separation, sparse labelling, and extreme class imbalance. In such cases, choosing a value of $\alpha$ close to the proportion of labelled data ($\ell$) often yields good predictive performance, suggesting that this heuristic may serve as a useful starting point. However, the relationship between the observed data and the optimal value of $\alpha$ is complicated and unpredictable, so we recommend empirical validation (e.g., cross validation over $\alpha$) in practice.

We used our weighted likelihood approach to effectively leverage underwater video and audio data and generate a more detailed description of killer whale foraging behaviour. Using cross validation, we showed that our weighted approach matches or (more commonly) outperforms the accuracy of previous baselines and single-frame machine learning methods. In addition to better performance, our method has a significant benefit over the baselines in that it infers an underlying temporal process that generates the time series itself. In this way, PHMMs allow researchers to understand latent behaviours in addition to categorizing them. The flexible structure of an HMM can also be extended in many ways, including to infer behaviours on multiple scales [7] or incorporate habitat covariates [57]. For example, in our second case study we used the structure of the PHMM to simultaneously infer fine-scale dive phases as well as successful foraging dives. We applied this approach to killer whales, but future work can focus on applying our methodology to other marine animals. We use a dive depth threshold of 30 metres to isolate killer whale foraging dives, but other marine animals may require different criteria to isolate foraging behaviour.

As is common in movement ecology, our case studies had a small number of labels, which can negatively impact the reliability of cross validation. Therefore, future work can apply this weighted likelihood approach to case studies with more labels, potentially from other fields. We also performed case studies in which we were confident in our labels. Future work can focus on how including labels affects the performance of the PHMM when researchers are less confident in their labels and infer $\beta$ as a parameter of the PHMM.

Although we did not have a large enough sample size or long enough tag deployments to draw definitive conclusions, the findings from our method are in line with those from Tennessen et al. [15], who conducted a comprehensive study of foraging behaviour in northern and southern resident killer whales. In particular, our results support their findings that northern resident killer whales tend to spend more time travelling and resting compared to southern residents, and that the northern residents tend to catch more fish per unit of effort compared to the southern residents.

Our weighted likelihood approach improves the estimates of unobserved labels from a time series, but it cannot ensure the interpretability and reliability of the labels themselves. For example, recent studies have found adult Chinook salmon migrating in the upper 30 metres of the water column in addition to the deep depths that we explored in this paper [58]. This suggests that killer whales likely employ two foraging strategies to exploit the dichotomy of Chinook swimming tactics. The first strategy that we and others have documented [15,59] involves repeatedly diving to depths exceeding 30 m to locate and pursue Chinook in areas where salmon are holding or evading predators. The other may involve killer whales repeatedly searching medium water depths (10–30 m) while travelling using echolocation to identify and capture salmon near the surface or by pursuing them to depth. As such, the foraging labels from the first case study likely correspond only to deep foraging dives associated

with this first foraging strategy. While we are confident that the PHMM accurately identified resting and foraging dives, we recognize that the killer whales may have been searching for Chinook during 'travelling' dives. In fact, we occasionally found echolocation clicks that coincided with dives labelled as travelling in our case study. It may thus be possible to further refine and validate the PHMM using echolocation data recorded by the tags to better categorize the behavioural states of killer whale travelling dives based on their depths and durations relative to the recent information on adult Chinook migratory behaviour.

We primarily focused on identifying the behaviour of killer whales, but incorporating sparse labels into complex HMMs is a common modelling problem across a variety of use cases and disciplines. In addition, complicated time series data are increasingly common as sensing technology continues to improve [8]. As such, the modelling approach developed here can help researchers effectively model complicated, sparsely labelled time series to optimize prediction accuracy and model fit.

## Supporting information

**S1 Appendix. Additional results from case study 1.** Figures displaying results from PHMMs fit using all five values of $\alpha$.
(PDF)

**S2 Appendix. Plots of data used in case and simulation studies.** Figures displaying scatter plots of data used in the case studies and the simulation study.
(PDF)

## Acknowledgments

We thank Mike deRoos and Chris Hall for assistance in the field with tag deployments, Taryn Scarff for assistance with drone deployments, the M/V Gikumi captain and crew, and Keith Holmes for piloting the drone, filming the killer whales, and assisting in synchronizing time stamps. Drone footage was collected in partnership with Hakai Institute. This research was enabled in part by support provided by WestGrid (www.westgrid.ca) and Compute Canada (www.computecanada.ca).

## Author contributions

**Conceptualization:** Evan Sidrow, Nancy Heckman, Marie Auger-Méthé.

**Data curation:** Evan Sidrow, Tess M McRae, Beth L. Volpov, Andrew W. Trites, Sarah M.E. Fortune.

**Formal analysis:** Evan Sidrow, Nancy Heckman, Marie Auger-Méthé.

**Funding acquisition:** Nancy Heckman, Andrew W. Trites, Sarah M.E. Fortune, Marie Auger-Méthé.

**Investigation:** Evan Sidrow, Nancy Heckman, Tess M McRae, Beth L. Volpov, Andrew W. Trites, Sarah M.E. Fortune, Marie Auger-Méthé.

**Methodology:** Evan Sidrow, Nancy Heckman, Tess M McRae, Beth L. Volpov, Marie Auger-Méthé.

**Project administration:** Nancy Heckman, Andrew W. Trites, Marie Auger-Méthé.

**Resources:** Nancy Heckman, Andrew W. Trites, Marie Auger-Méthé.

**Software:** Evan Sidrow.

**Supervision:** Nancy Heckman, Beth L. Volpov, Andrew W. Trites, Sarah M.E. Fortune, Marie Auger-Méthé.

**Validation:** Nancy Heckman, Tess M McRae, Beth L. Volpov, Andrew W. Trites, Sarah M.E. Fortune, Marie Auger-Méthé.

**Visualization:** Evan Sidrow.

**Writing – original draft:** Evan Sidrow.

**Writing – review & editing:** Nancy Heckman, Tess M McRae, Beth L. Volpov, Andrew W. Trites, Sarah M.E. Fortune, Marie Auger-Méthé.

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
