## [Decision Letter · Decision Letter 0]

15 Dec 2024

PONE-D-24-46828Incorporating sparse labels into hidden Markov models using weighted likelihoods improves accuracy and interpretability in biologging studiesPLOS ONE

Dear Dr. Sidrow,

Thank you for submitting your manuscript to PLOS ONE. After careful consideration, we feel that it has merit but does not fully meet PLOS ONE’s publication criteria as it currently stands. Therefore, we invite you to submit a revised version of the manuscript that addresses the points raised during the review process.

We look forward to receiving your revised manuscript.

Kind regards,

Vitor Hugo Rodrigues Paiva, Ph.D.

Academic Editor

PLOS ONE

Journal Requirements:

5. We note you have included a table to which you do not refer in the text of your manuscript. Please ensure that you refer to Table 2 in your text; if accepted, production will need this reference to link the reader to the Table.

Reviewers' comments:

Reviewer's Responses to Questions

**Comments to the Author**

1. Is the manuscript technically sound, and do the data support the conclusions?

Reviewer #1: Partly

Reviewer #2: Yes

2. Has the statistical analysis been performed appropriately and rigorously? 

Reviewer #1: Yes

Reviewer #2: Yes

3. Have the authors made all data underlying the findings in their manuscript fully available?

Reviewer #1: Yes

Reviewer #2: Yes

4. Is the manuscript presented in an intelligible fashion and written in standard English?

Reviewer #1: Yes

Reviewer #2: Yes

5. Review Comments to the Author

Reviewer #1: This study introduces a weighted likelihood approach for hidden Markov models (HMMs) to address the challenge of sparse labeling in ecological time series data. Through case studies on killer whale foraging behavior, the approach demonstrates notable improvements in accuracy and interpretability compared to existing methods. Overall, the manuscript is well-organized and effectively presented. However, several enhancements could further strengthen its completeness and rigor.

Data Visualization: It would be highly beneficial to include plots that showcase the actual distribution of the data points. For Case Study 1, a 2D scatter plot of maximum depth ( m ) against dive duration ( d ), color-coded by dive types, would provide an intuitive visualization of the data. Similarly, for Case Study 2, a 3D scatter plot of depth change (d) , heading variation ( h ), and jerk peak ( j ), color-coded by labels, would help illustrate the feature space and label distribution.

Comparison with Single-Frame Methods: Please perform a side-by-side comparison of the proposed HMM-based methods (with varying \alpha ) against single-frame methods, such as random forest, SVM, etc. Intuitively, the temporal information utilized by multi-frame HMMs should lead to better performance than single-frame methods, while the proposed weighted likelihood approach is expected to outperform standard HMMs due to the inclusion of label weighting. Quantifying these performance improvements in a clear and systematic manner would provide strong support for the method’s efficacy.

Equation and Figure Clarifications: In Equation (2), the parameter \delta should be explicitly defined for clarity. Additionally, in Figure 2, it is difficult to distinguish between rows A, B, and C. Enhancing the figure with more distinct visual markers or annotations would improve its readability and interpretation.

Reviewer #2: This paper provides a detailed description of a novel statistical method for incorporating partially labelled data in hidden Markov models to estimate more detailed state processes in animal movement studies. This paper functions both as a methodology paper, and does a good job of describing the statistical theory and methods, as well as providing novel ecological results in the case studies about killer whale foraging behavior. While I appreciate the ecological relevance of the killer whale case studies, from a methodological point of view I would have loved to see an additional case study with a comparison to a case where there is more labelled data, or the data labelling comes from multiple processes. I think this would help the reader better understand when they should choose to use this method. Additionally, I was confused about the way the cross-validation procedure worked, especially in terms of using individual killer whale data but in different folds. I suggest that the authors rewrite these sections to better clarify this procedure, as well as to perhaps include a conceptual diagram. Overall, I believe this paper is a valuable contribution to the field and should be published.

Minor comments:

Line 15: re-word “what animals are doing”, perhaps: “fine-scale animal behavior”

Line 18: Can you be more specific about what “various settings” means here?

Line 45-46: These paragraphs do not feel well connected, perhaps add a transition sentence between the two

Line 48: I believe “Northern” and “Southern” are traditionally capitalized in the context of killer whales

Lines 68 - 74: I think it would be useful somewhere in here to specify that states are discrete but that observations are usually continuous

Lines 67 - 86: I’m not sure you need the full description of HMMs here, as they are relatively commonly used these days. Perhaps this section can be shortened.

Line 129: Replace “throws out” with “removes”

Line 279: I noticed that in both case studies you match the state structure with the number of states in your labels (3 dive types, 3 states) what would happen if these did not match?

Lines 368-370: I understand the biological reason for limiting the data to dives > 30 m, but was there a statistical reason as well? How might this apply to other taxa?

Figure 3: I think if you remove the gray background to these plots it will help make the gray bars more distinguishable

6. PLOS authors have the option to publish the peer review history of their article (what does this mean?). If published, this will include your full peer review and any attached files.

Reviewer #1: No

Reviewer #2: No

---

## [Author Response · Author response to Decision Letter 1]

22 Apr 2025

Journal Requirements:

1. Please ensure that your manuscript meets PLOS ONE's style requirements, including those for file naming. The PLOS ONE style templates can be found at  https://journals.plos.org/plosone/s/file?id=wjVg/PLOSOne_formatting_sample_main_body.pdf and  https://journals.plos.org/plosone/s/file?id=ba62/PLOSOne_formatting_sample_title_authors_affiliations.pdf

We have read the style templates above to ensure that our manuscript meets PLOS ONE’s style requirements.

All of our code is available at https://github.com/EvanSidrow/PHMM.

We have added the following statement at the beginning of our “Case studies” section to provide additional information about our permits.

The data for these case studies were collected with written approval under the University of British Columbia Animal Care Permit no. A19-0053, Fisheries and Oceans Canada Marine Mammal Scientific License for Whale Research no. XMMS 6 2020, and United States Department of Commerce, NOAA, National Marine Fisheries Service Permit No. 23220. Collection took place in August and September 2020 off the coast of British Columbia in Queen Charlotte Sound, Queen Charlotte Strait, Johnstone Strait, and Juan de Fuca Strait.

We believe that our statement above at the beginning of our “Case studies” addresses this point, but please let us know if anything else is needed.

5. We note you have included a table to which you do not refer in the text of your manuscript. Please ensure that you refer to Table 2 in your text; if accepted, production will need this reference to link the reader to the Table.

To the best of our knowledge we have no tables in our manuscript. We had referred to Table 2 of McRae et al. (2024), but we did not explicitly show this table. We have removed the reference to Table 2, and instead simply reference McRae et al. (2024) to avoid confusion.   

Reviewer #1: This study introduces a weighted likelihood approach for hidden Markov models (HMMs) to address the challenge of sparse labeling in ecological time series data. Through case studies on killer whale foraging behavior, the approach demonstrates notable improvements in accuracy and interpretability compared to existing methods. Overall, the manuscript is well-organized and effectively presented. However, several enhancements could further strengthen its completeness and rigor.

Thank you for the kind words and constructive comments. We believe the changes we made in response to the comments further strengthen the manuscript.

Data Visualization: It would be highly beneficial to include plots that showcase the actual distribution of the data points. For Case Study 1, a 2D scatter plot of maximum depth ( m ) against dive duration ( d ), color-coded by dive types, would provide an intuitive visualization of the data. Similarly, for Case Study 2, a 3D scatter plot of depth change (d) , heading variation ( h ), and jerk peak ( j ), color-coded by labels, would help illustrate the feature space and label distribution.

We agree that scatter plots improve the interpretability of our case studies. We have added scatter plots that show the distribution of the data points for case study 1 (Fig S3) and case study 2 (Fig S4 and Fig S5). We have also plotted an example data set from our new simulation study (Fig S6). All figures are in Appendix S2.

 Comparison with Single-Frame Methods: Please perform a side-by-side comparison of the proposed HMM-based methods (with varying \alpha ) against single-frame methods, such as random forest, SVM, etc. Intuitively, the temporal information utilized by multi-frame HMMs should lead to better performance than single-frame methods, while the proposed weighted likelihood approach is expected to outperform standard HMMs due to the inclusion of label weighting. Quantifying these performance improvements in a clear and systematic manner would provide strong support for the method’s efficacy.

Thank you for this suggestion. We have added three single-frame methods to case studies 1 and 2: logistic regression, random forests, and SVMs (See line number 339-342). The PHMMs outperform the single frame methods in case study 1, but are matched by single frame methods in case study 2. We believe this is because the labelled foraging dives are well-separated based on average bottom heading variation and jerk peak (see our new Fig S5). Nonetheless, we believe that our method is still useful for case study 2 because it is as accurate as the single frame methods but simultaneously categorizes dive phases as well as successful foraging dives. It also infers the underlying generative process for the killer whale data.

 

Equation and Figure Clarifications: In Equation (2), the parameter \delta should be explicitly defined for clarity. Additionally, in Figure 2, it is difficult to distinguish between rows A, B, and C. Enhancing the figure with more distinct visual markers or annotations would improve its readability and interpretation.

We have added an Equation to define delta more clearly (now Equation (1)), and have added additional wording prior to Equation (2) (now Equation (3)) to clarify \delta (lines 84-86). We have also adjusted Figure 2 to distinguish more clearly between rows A, B, and C.

Reviewer #2: This paper provides a detailed description of a novel statistical method for incorporating partially labelled data in hidden Markov models to estimate more detailed state processes in animal movement studies. This paper functions both as a methodology paper, and does a good job of describing the statistical theory and methods, as well as providing novel ecological results in the case studies about killer whale foraging behavior.

Thank you for the kind words and constructive comments. The changes we made in response to them have improved the manuscript.

While I appreciate the ecological relevance of the killer whale case studies, from a methodological point of view I would have loved to see an additional case study with a comparison to a case where there is more labelled data, or the data labelling comes from multiple processes. I think this would help the reader better understand when they should choose to use this method.

While a new case study would add to the manuscript, we believe that a simulation study would better demonstrate when the method is most useful in a controlled setting, which we understood as the main comment here. As such, we have added a simulation study after the case studies to investigate the effect of label density on the ideal value of alpha. The simulation study also investigates the effects of model misspecification, state separation, data set size, and class imbalance. We determined that all of these factors can affect the ideal value of alpha. In summary, we conclude that our method is useful in difficult modelling scenarios (fewer labels, more model misspecification, less state separation, etc.). However, it is difficult to know in practice how the data affects the ideal value of alpha, so we recommend using our approach with cross-validation to find the ideal value of alpha. Also note that our method is a generalization of the standard HMM approach. (lines 563-634).

Additionally, I was confused about the way the cross-validation procedure worked, especially in terms of using individual killer whale data but in different folds. I suggest that the authors rewrite these sections to better clarify this procedure, as well as to perhaps include a conceptual diagram. Overall, I believe this paper is a valuable contribution to the field and should be published.

We have re-worded the cross-validation procedure (lines 207-231) and added a conceptual diagram (Fig 3).

Minor comments: Line 15: re-word “what animals are doing”, perhaps: “fine-scale animal behavior”

We have made the suggested change (line 26).

Line 18: Can you be more specific about what “various settings” means here?

We have changed the wording in this sentence to clarify (line 29).

Line 45-46: These paragraphs do not feel well connected, perhaps add a transition sentence between the two

We agree that the paragraph is out of place. As such, we have moved the second paragraph earlier in the introduction to make the flow of the introduction better (lines 15-25).

Line 48: I believe “Northern” and “Southern” are traditionally capitalized in the context of killer whales

We have double checked the literature and have found mixed results: some papers use upper-case and some use lower-case. However, we have found that “northern” and “southern” generally tend to be lower-case. Some examples include:

https://www.sararegistry.gc.ca/document/doc1341a/p1_e.cfm

https://onlinelibrary.wiley.com/doi/epdf/10.1111/gcb.17490

https://cdnsciencepub.com/doi/10.1139/cjfas-2020-0445

Lines 68 - 74: I think it would be useful somewhere in here to specify that states are discrete but that observations are usually continuous

We have made this change in the manuscript (lines 72-74).

Lines 67 - 86: I’m not sure you need the full description of HMMs here, as they are relatively commonly used these days. Perhaps this section can be shortened.

While we agree that HMMs are relatively common and that most readers can skip this subsection, we had trouble balancing this request with one from another reviewer to expand on the section. The section also introduces notation that is used later in the manuscript.

Line 129: Replace “throws out” with “removes”

We have made this change in the manuscript (line 135).

Line 279: I noticed that in both case studies you match the state structure with the number of states in your labels (3 dive types, 3 states) what would happen if these did not match?

This is an important observation, and this phenomenon can cause complications in the model. In fact, in the second case study we do not match the state structure with the number of distinct label types. If there are more states in the HMM than there are distinct label types, then it is not advisable to set $\alpha = 0$, since then there would be no information to estimate any unlabelled hidden states. We mentioned this problem briefly in the second case study, but we have moved the explanation of the issue up to the “model” section because it is an important consideration for all PHMMs. In addition, if a single label could correspond to multiple components (e.g. there are two kinds of foraging but a foraging label cannot distinguish between the two), then we recommend parameterizing g^{(i)}. We have clarified this on lines 126 as well as lines 213-216.

Lines 368-370: I understand the biological reason for limiting the data to dives > 30 m, but was there a statistical reason as well? How might this apply to other taxa?

From a statistical standpoint, the HMM from the second case study is likely a bad model for dives < 30m because heading total variation, jerk peak, etc. is likely distributed differently for shallow dives compared to deep foraging dives. We have added this information to case study 2 in lines 412-414. In addition, we believe that our modelling approach can work for other taxa, but dividing up foraging and non-foraging dives should be made on a case-by-case basis. We have added this to the discussion section of the manuscript in lines 663-666.

Figure 3: I think if you remove the gray background to these plots it will help make the gray bars more distinguishable

Thank you for the suggestion- we have made the changes in the manuscript (now Figure 4).

---

## [Decision Letter · Decision Letter 1]

12 May 2025

Incorporating sparse labels into hidden Markov models using weighted likelihoods improves accuracy and interpretability in biologging studies

PONE-D-24-46828R1

Dear Dr. Sidrow,

We’re pleased to inform you that your manuscript has been judged scientifically suitable for publication and will be formally accepted for publication once it meets all outstanding technical requirements.

Kind regards,

Vitor Hugo Rodrigues Paiva, Ph.D.

Academic Editor

PLOS ONE

Additional Editor Comments (optional):

Reviewers' comments:

Reviewer's Responses to Questions

**Comments to the Author**

1. If the authors have adequately addressed your comments raised in a previous round of review and you feel that this manuscript is now acceptable for publication, you may indicate that here to bypass the “Comments to the Author” section, enter your conflict of interest statement in the “Confidential to Editor” section, and submit your "Accept" recommendation.

Reviewer #1: All comments have been addressed

Reviewer #2: All comments have been addressed

2. Is the manuscript technically sound, and do the data support the conclusions?

Reviewer #1: Yes

Reviewer #2: Yes

3. Has the statistical analysis been performed appropriately and rigorously? 

Reviewer #1: Yes

Reviewer #2: Yes

4. Have the authors made all data underlying the findings in their manuscript fully available?

Reviewer #1: Yes

Reviewer #2: Yes

5. Is the manuscript presented in an intelligible fashion and written in standard English?

Reviewer #1: Yes

Reviewer #2: Yes

6. Review Comments to the Author

Reviewer #1: (No Response)

Reviewer #2: The revision has greatly improved the manuscript and I believe it should be accepted for publication.

7. PLOS authors have the option to publish the peer review history of their article (what does this mean?). If published, this will include your full peer review and any attached files.

Reviewer #1: No

Reviewer #2: No

---

## [Editor Report · Acceptance letter]

PONE-D-24-46828R1

PLOS ONE

Dear Dr. Sidrow,

I'm pleased to inform you that your manuscript has been deemed suitable for publication in PLOS ONE. Congratulations! Your manuscript is now being handed over to our production team.

Kind regards,

on behalf of

Dr. Vitor Hugo Rodrigues Paiva

Academic Editor

PLOS ONE